# A Novel Antimicrobial Mechanism of Azalomycin F Acting on Lipoteichoic Acid Synthase and Cell Envelope

**DOI:** 10.3390/molecules29040856

**Published:** 2024-02-14

**Authors:** Zilong Luo, Pingyi Li, Duoduo Zhang, Jianping Zhu, Wan Wang, Wenjia Zhao, Peibo Li, Ganjun Yuan

**Affiliations:** 1Biotechnological Engineering Center for Pharmaceutical Research and Development, Jiangxi Agricultural University, Nanchang 330045, China; 2Laboratory of Natural Medicine and Microbiological Drug, College of Bioscience and Bioengineering, Jiangxi Agricultural University, Nanchang 330045, China; 3School of Basic Medicine, Nanchang Medical College, Nanchang 330006, China; 4School of Life Sciences, Sun Yat-sen University, Guangzhou 510275, China

**Keywords:** azalomycin F, *Staphylococcus aureus*, lipoteichoic acid, fluorescence, molecular dock, virulence factor, proteomics

## Abstract

Lipoteichoic acid (LTA) plays an essential role in bacterial growth and resistance to antibiotics, and LTA synthetase (LtaS) was considered as an attractive target for combating Gram-positive infections. Azalomycin F, a natural guanidyl-containing polyhydroxy macrolide, can target the LTA of *Staphylococcus aureus*. Using various technologies including enzyme-linked immunosorbent assay, transmission electron microscope, proteomics, and parallel reaction monitoring, here, the experimental results indicated that azalomycin F can accelerate the LTA release and disrupt the cell envelope, which would also lead to the feedback upregulation on the expressions of LtaS and other related enzymes. Simultaneously, the reconstituted enzyme activity evaluations showed that azalomycin F can significantly inhibit the extracellular catalytic domain of LtaS (eLtaS), while this was vague for LtaS embedded in the liposomes. Subsequently, the fluorescence analyses for five incubation systems containing azalomycin F and eLtaS or the LtaS-embedded liposome indicated that azalomcyin F can spontaneously bind to the active center of LtaS. Combining the mass spectroscopy analyses and the molecular dockings, the results further indicated that this interaction involves the binding sites of substrates and the LTA prolongation, especially the residues Lys299, Phe353, Trp354 and His416. All these suggested that azalomycin F has multiple antibacterial mechanisms against *S. aureus*. It can not only inhibit LTA biosynthesis through the interactions of its guanidyl side chain with the active center of LtaS but also disrupt the cell envelope through the synergistic effect of accelerating the LTA release, damaging the cell membrane, and electrostatically interacting with LTA. Simultaneously, these antibacterial mechanisms exhibit a synergistic inhibition effect on *S. aureus* cells, which would eventually cause the cellular autolysis.

## 1. Introduction

Antimicrobial-resistant pathogens have been emerging as a global threat to human health [1], and the COVID-19 pandemic has further accelerated this global problem [2]. Among them, *Staphylococcus aureus*, as a representative of Gram-positive bacteria, can cause mild to serious infections including skin and soft-tissue ones, endocarditis, osteomyelitis, and meningitis [3], and therefore there is a desperate need for new antimicrobial agents and efficacious therapeutics [4,5].

Since many virulence factors simultaneously involve the bacterial pathogenicity to human body and the evasion from the host immune system, the anti-virulence factor is considered as an especially compelling alternative strategy due to its low selective pressure for the development of drug resistance in bacteria [6]. Among the virulence factors of *S. aureus*, lipoteichoic acid (LTA), an anionic surface polymer anchoring to the cell membrane and consisting of glycerol phosphate repeats [7,8], plays an important role in bacterial cell physiology and virulence, such as bacterial growth, cell division, biofilm formation, autolysin regulation, and resistance to cationic antibiotics [7,8]. Thereby, its synthase LtaS has emerged as an attractive target of antibacterial drugs combating staphylococcal infections [9,10].

So far, only three lipoteichoic acid synthase (LtaS) inhibitors were reported: compound 1771 (2-oxo-2-(5-phenyl-1,3,4-oxadiazol-2-ylamino) ethyl 2-naphtho[2,1-b]furan-1-ylacetate) [10], the dye Congo red [8] and compound 4 (2-oxo-2-(5-phenyl-1,3,4-oxadiazol-2-ylamino) ethyl 1-phenyl-3-(2-thienyl)-1*H*-pyrazole-4-carboxylate) [9]. Based on the structure of compound 1771, compound 4 was designed and showed greater affinity for binding to the extracellular catalytic domain of LtaS (eLtaS) [9]. However, Vickery et al. had confirmed that compound 1771 did not inhibit LtaS directly and suggested that it probably inhibited another enzyme required for polymer production and had a more general toxicity [8]. This was also supported by the conclusion reported by Douglas et al. based on the critical facts that neither the deletion nor overexpression of LtaS alter the susceptibility of *S. aureus* to compounds 1771 and its derivative 13, while the overexpression of LtaS can alter that to Congo red [11]. Muscato et al. discovered two inhibitors of the glycosyltransferase UgtP, which assembles the LTA glycolipid anchoring on the *S. aureus* cell membrane [12]. Regretfully, the carcinogenicity of the sole LtaS inhibitor Congo red limits its potential as an antibiotic [8,9]. So, it is very necessary to discover novel LtaS inhibitors with more potency for the development of antimicrobial agents targeting the LTA.

Azalomycin F, a natural guanidyl-containing polyhydroxy macrolide produced by some streptomycete strains, has various bioactivities against Gram-positive bacteria, yeast, fungi, etc. [13]. It contains three main components, azalomycins F_3a_, F_4a_ and F_5a_ (Figure 1) [13,14,15], and their relative configurations were reported in 2013 [16]. Based on the chemical and genomic analyses, the absolute configurations of their analogs niphimycins were proposed in 2018 [17], and their absolute configurations were further suggested in 2021 [18]. Simultaneously, the antimicrobial mechanism showed that azalomyin F can disrupt the cell membrane of *S. aureus* through the synergistic effects of its lactone ring binding to the polar head of the cell-membrane phospholipid and its guanidyl-containing side chain targeting the LTA [19]. These results showed a good research value for azalomycin F as a novel antimicrobial agent. However, it remained unknown how azalomycin F targets *S. aureus* LTA, such as inhibiting LtaS [8], influencing the function of LTA, and/or accelerating the release of LTA [20]. Thereby, here, the detailed mechanisms were further explored for azalomycin F targeting *S. aureus* LTA, according to Figure 1.

## 2. Results and Discussion

### 2.1. Azalomycin F Accelerating the Release of Cellular LTA

The influences of azalomycin F on the release of cellular LTA from *S. aureus* ATCC 25923 were evaluated, and the results (Figure 2a) showed that there is no significant difference between the azalomycin F group (1 or 2 μg/mL) and blank group (0 μg/mL) in 150 min. Differently, it can be observed that both 4 and 8 μg/mL azalomycin F groups obviously increased the LTA release. However, significant differences relative to the blank group were only presented for the azalomycin F group (8 μg/mL) in 90 min and for the azalomycin F group (4 μg/mL) in 30 min. This might be attributed to their large experimental errors for other timepoints and the larger initial values of blank groups relative to all azalomycin F groups. The latter was also confirmed, as the LTA releases of both the 4 and 8 μg/mL azalomycin F groups were significantly larger than those of the 1 and 2 μg/mL azalomycin F ones. Therefore, these above indicated overall that the LTA release of *S. aureus* remarkably increased when the interfering concentrations of azalomycin F were more than or equal to its minimum inhibitory concentration (MIC) of 4 μg/mL. Simultaneously, a non-concentration-dependent release of LTA after being treated with 4 and 8 µg/mL azalomycin F indicated that azalomycin F does not have a direct relation to promoting the LTA release. This might be attributed to the simultaneous membrane damage of azalomycin F to *S. aureus* [19].

Since our previous work concluded that azalomycin F can disrupt the cell membrane of *S. aureus* [19], it seems that the LTA release of *S. aureus* would definitely increase when the cell membrane was damaged. To explore whether the increase in LTA release is completely due to the membrane damage, here, the conclusion on membrane damage was more intuitively proved by the observation on the ultrathin slices of *S. aureus* cells after being treated with azalomycin F. Figure 2b shows that some parts of the cell membrane were damaged, and the leakages of cellular substances increased along with the concentration increase in azalomycin F interfering with *S. aureus* growth. Moreover, some parts of the boundary between the cell wall and the cell membrane were unclear, and the cell wall at those clear boundaries thickened. As the loss of LTA would lead to the release of cell autolysins and subsequently the destabilization of the bacterial cell wall [21], the latter was consistent with the feedback upregulation for some proteins related to cell wall biosynthesis that resulted from subsequent proteome experiments, and it also coincided with the previous report [10]. It was worth noting that a clear separation of the cell membrane and cell wall of *S. aureus* after being treated with 8 μg/mL azalomycin F was observed in Figure 2b. Since LTA anchors to the cell membrane and binds to the peptidoglycan, this might be also due to the decrease in LTA with normal physiological activity [21]. Like some cationic antimicrobial peptides [22], probably the electrostatic interaction between the guanidyl group of azamycin F and the negatively charged phosphate of LTA would also accelerate the LTA release, as both groups are close to each other when the lactone ring of azalomycin F bound to the polar head of the cell membrane phospholipid [19]. Thereby, these indicated that azalomycin F can independently accelerate the LTA release of *S. aureus* separate from that resulting from membrane damage. Simultaneously, both the promotion of LTA release and the membrane damage were mutually reinforcing for azalomycin F against *S. aureus*. Furthermore, using a biochemical assay, the kinetic experiment for membrane integrity along with LTA release will shed more light on the relationship between LTA release and membrane damage.

Similar to some β-lactam antibiotic and ciprofloxacin [20,23], azalomycin F can increase the LTA release of S. aureus. However, they likely have different molecular mechanisms. The accelerating release of S. aureus LTA was not only due to the damage to the cell membrane but also achieved by the probable binding of azalomycin F to the phosphodiester anionic of LTA through electrostatic force [22,24], especially at the position close to the initial position of LTA extension or near the cell membrane. It may be that azalomycin F could also interfere with the physiological function of LTA, as the positive charged guanidyl side chain generally lies outside of the cell membrane [19].

### 2.2. Influence of Azalomycin F on the Expressions of LtaS and Other Related Enzymes

LC-MS analyses indicated that 1709 quantifiable proteins were identified from the proteome of *S. aureus* treated with and without azalomycin F (Appendix A). According to the thresholds (1.2 of fold change) of significant changes (*p* < 0.05) for differential expression levels, 346 of them were upregulated, and 342 of them were downregulated (Appendix A). The functional enrichment analysis for Gene Ontology (GO) (Figure 3a,b) and Kyoto Encyclopedia of Genes and Genomes (KEGG) (Figure 3c) indicated that a series of biological processes and key pathways, including the cell wall macromolecule metabolic process (especially peptidoglycan biosynthesis; see Appendix A), the regulation of cell shape and morphogenesis, and *S. aureus* infection, were significantly affected by azalomycin F, and they also involved many cellular components located in the periplasmic space, cell envelope and extracellular region of *S. aureus*. Heatmaps of cluster analyses for the GO in biological processes (Figure 3d) and the KEGG pathway (Figure 3f) indicated that many proteins involving various biological processes of the cell envelope, the regulation of cell shape and morphogenesis, and the syntheses of peptidoglycan were significantly upregulated, and this coincided with the thickening cell wall observed from Figure 2b. Simultaneously, those used for the GO functional enrichment (Figure 3e) showed that some proteins related to the cellular components of the cell envelope and extracellular region were downregulated.

Among those differential expression proteins, it was noteworthy that the LTA synthase (LtaS) expression was upregulated about 1.27 times. Therefore, LtaS together with another five proteins related to the LTA function, the syntheses of the cell wall and the stabilization of the cell envelope were selected for further verifiably quantification using the parallel reaction monitoring (PRM) technology. The results showed that three proteins, lipoteichoic acid synthase (Q2G093), signal peptidase I (Q2FZT7) and teichoic acids export ATP-binding protein TagH (Q2G2L1), related to the LTA biosynthesis and function, were upregulated 1.23 (Appendix A), 1.19 (Appendix A), and 1.48 (Appendix A) times, respectively. However, lipoprotein (Q2G0V0), a protein covalently connected to the tetrapeptide side chain of peptidoglycan through its protein part at one end and to the phosphoric acid of the membrane through its lipid part at the other end, was downregulated to 32% of that in blank group (Appendix A). Moreover, both UDP-*N*-acetylmuramoyl-tripeptide-D-alanyl-D-alanine ligase (Q2FWH4) and D-alanine-D-alanine ligase (Q2FWH3), involving the cell wall synthesis of *S. aureus*, were, respectively, upregulated 1.42 (Appendix A) and 1.47 (Appendix A) times, and this was also consistent with the observed fact (Figure 2b) that the cell wall of *S. aureus* thickened after being treated with azalomycin F. Thereby, these above proposed that azalomycin F can lead to the significant upregulations for the expressions of LtaS and other proteins related to the LTA function, the cell wall synthesis and the stabilization of the cell envelope.

Since LTA plays an essential role in bacterial growth, cell division, biofilm formation, autolysin regulation and resistance to cationic antibiotics [8,9,25], azalomycin F increasing the LTA release, together with inhibiting the LTA biosynthesis (subsequently concluded in Section 2.3), would theoretically lead to the feedback increase in related enzyme expression, addressing the disadvantageous effects of LTA deficiency in bacterial cells. This was just consistent with the upregulations of some proteins (such as LtaS, signal peptidase I, and teichoic acids exporting ATP-binding protein TagH) related to the biosynthesis and transport of LTA after *S. aureus* was treated with azalomycin F. Furthermore, this was also supported from the upregulations of some proteins involving the biosynthesis of some components for stabilizing the cell envelope, such as UDP-*N*-acetylmuramoyl-tripeptide-D-alanyl-D-alanine ligase and D-alanine-D-alanine ligase.

Thereby, from the perspective of biofeedback, these above together with the electron micrographs in Figure 2b confirmed that azalomycin F can remarkably disrupt the bacterial cell envelope and inhibit the syntheses of some important cellular components in the bacterial cell envelope.

### 2.3. Azalomycin F Inhibiting LtaS to Synthesize LTA

To evaluate the influence of azalomycin F on LtaS synthesizing LTA, two enzymes eLtaS (49.3 kDa) and LtaS (74.4 kDa) were expressed in *E. coli* BL21(DE3)/pET28a(+)-eLtaS and -LtaS, and they were verified using SDS–PAGE (Figure 4a,b) and Western blot technologies (Figure 4c,d). After being purified by the Ni^2+^-NTA-Sepharose affinity column, and desalted and concentrated by an ultrafiltration centrifuge tube, the concentrations of both two enzymes were determined. Generally, the concentrations of eLtaS prepared would fall into the range from 3 to 7 mg/mL. However, the prepared LtaS generally contained about two times the proportion of eLtaS, and their concentrations were, respectively, 72 μg/mL and 168 μg/mL, since the membrane protein LtaS can be usually degraded into eLtaS during the processes of its expression, isolation and purification. Using L-*α*-phosphatidyl-DL-glycerol (sodium salt) (PG) as the substrate, the evaluation on enzyme activities indicated that the purified eLtaS and LtaS, together with the subsequent LtaS-embedded liposome, can well catalyze PG to produce diacylglycerol (DAG) and presented good enzymatic activities (Appendix A). Coinciding with a previous report [25], the concentration of Mn^2+^ in the incubation system can also affect the enzymatic reaction and the content of the product DAG.

Using PG as the substrate, the influences of azalomycin F on eLtaS synthesizing LTA were evaluated from the incubation system of eLtaS/PG/azalomycin F. Overall, the results (Figure 4e) indicated that the enzyme activity of eLtaS can be obviously inhibited by azalomycin F with concentrations ranging from 20 to 640 μM. Due to the large relative standard deviation, azalomycin F presented significant (*p* < 0.05) inhibition to eLtaS only for both groups as 80 and 640 μM, being relative to the blank group (0 μM). In another study, the inhibition of 640 μM azalomycin F was significantly (*p* < 0.05) larger than that of 80 μM azalomycin F to eLtaS, while there was no significant difference among the other five groups with azalomycin F concentrations ranging from 20 to 320 μM. This might be attributed to the large intra-group errors in the experiments. Thereby, it was confirmed that azalomycin F can inhibit the LTA biosynthesis from eLtaS, although only 80 and 640 μM azalomycin F groups, respectively, presented significant and very significant difference in statistics. More importantly, it was further confirmed by the fact that azalomycin F-induced *S. aureus* lysis can be can prevented by the cellular LTA [19]. This evidence is very crucial for the judgement of whether a compound is an LtaS inhibitor or not, since some compounds like compound 1171 and its derivatives were eventually proved to be not LtaS inhibitors except for Congo red, which was mainly because neither the deletion nor overexpression of LtaS altered the susceptibility of *S. aureus* to them [11].

Since LtaS is a transmembrane protein, the LtaS-embedded liposome was further prepared to simulate the physiological environment of the LTA biosynthesis in *S. aureus* as far as possible, using the cell membrane phospholipids extracted from *S. aureus* according to the reported method [26] and analyzed by thin-layer chromatography (TLC) (Figure 4f,g). Its average size diameter with a polydisperse index (PDI) of 0.30 was 262.9 nm (Figure 4h), which was larger than that of LUVs (88.8 nm) with a PDI of 0.08, and its surface charge was −16.4 mv, which was greater than the zeta potential (−33.9 mv) of the liposomes without chimeric LtaS. These showed that LtaS was well embedded in the LUVs, and the LtaS-embedded liposome presented good uniformity and stability. Subsequently, the influences of azalomycin F on LtaS synthesizing LTA were further evaluated from the incubation system of LtaS-embedded liposome/PG/azalomycin F. Different from the incubation systems eLtaS/PG/azalomycin F, the enzyme activity of LtaS embedded in the liposomes was obviously enhanced by azalomycin F at concentrations of 20 μM (*p* < 0.05) or 80 μM (*p* < 0.01) (Figure 4i), being relative to blank group (0 μM). However, there were no significant differences (*p* > 0.05) among various concentrations ranging from 10 to 160 μM of azalomycin F. In another study, observed from Figure 4i, the LtaS enzyme activities presented a fluctuating change and no significant difference between the interfered concentration of azalomycin F at 0 and 160 μM. Thereby, it was not concluded from Figure 4i that azalomycin F can inhibit LtaS to synthesize LTA in the incubation system of an LtaS-embedded liposome/PG/azalomycin F. Considering the prevention of azalomycin F-induced *S. aureus* lysis by the cellular LTA [19] and the inhibition of azalomycin F on eLtaS to synthesize the LTA, this might be due to the actual concentrations of azalomycin F acting on the LtaS embedded in the liposome being largely reduced from the molecular consumption of azalomycin F binding to the phospholipid polar head of the liposome [19]. Nonetheless, the downward trend could be observed after the incubation system showed interference from the concentrations of azalomycin F from 80 to 160 μM. Simultaneously, during the preparation and incubation of the LtaS-embedded liposome, some residual LtaS unembedded and possible eLtaS degraded from LtaS might be also responsible for this result, since the eLtaS and LtaS unembedded in the liposome can also catalyze PG to synthesize the LTA.

As LTA is related to the regulation of cell shape and morphogenesis, as well as *S. aureus* infection, the above findings further confirmed, combined with a previous report [19], that azalomycin F can reduce LTA biosynthesis by inhibiting LtaS. Simultaneously, this can also give some reasonable interpretation for the feedback upregulated expressions of LtaS and other proteins related to the LTA function and the synthesis of cellular components in the cell envelope such as peptidoglycan.

### 2.4. Action Mode of Azalomycin F Interacting with LtaS

Since most proteins contain intrinsically fluorescent groups such as tryptophan and tyrosine residues, the fluorescent changes can conversely reflect the microenvironment changes around their chromophores [27], and therefore the fluorescence spectroscopy is widely used to study the interaction between drugs and proteins [28]. To understand the interactions of azalomycin F with LtaS, five incubation systems—(A) eLtaS/azalomycin F, (B) eLtaS/DPPG/azalomycin F, (C) eLtaS/DPPG/MnCl_2_/azalomycin F, (D) LtaS-embedded liposome/azalomycin F, and (E) LtaS-embedded liposome/MnCl_2_/azalomycin F—were selected for exploring the fluorescence quenching of LtaS by azalomycin F. It is worth noting that 1,2-dihexadecanoyl-*sn*-glycero-3-phospho-(1′-rac-glycerol) (DPPG) is a main component of cell membrane phospholipids of *S. aureus* and here was mainly used as a component of control systems (B) and (C) relative to systems (D) and (E), which contained LtaS-embedded liposomes. The fluorescence spectra with the excitation wavelength (*λ*_ex_) at 280 nm were recorded and shown in Figure 5. It indicated that the fluorescence intensities of all these five systems, whether at 25 °C or at 35 °C, gradually decreased along with the increase in azalomycin F concentration. This showed that azalomycin F can interact with the intrinsic fluorescent substance eLtaS and LtaS, respectively. Differently, the potency of azalomycin F quenching the intrinsic fluorescent substance at 25 °C was obviously greater than that at 35 °C for systems (A) to (C), but it was a little smaller for systems (D) and (E). Based on previous work [19], it was deduced that the strong interaction between azalomycin F and the phospholipids of the cell membrane would weaken its quenching of the intrinsic fluorescence of LtaS.

To further explore the quenching mechanisms, binding forces and probable cooperative effects of azalomycin F interacting with LtaS, the Stern–Volmer quenching constant (*K*_sv_), the quenching constant (*K*_q_), the binding constant (*K*_a_), the number of binding sites (*n*), various thermodynamic parameters including enthalpy change (Δ*H*), entropy change (Δ*S*) and Gibbs energy change (Δ*G*), and Hill’s coefficient (*n*_H_) were calculated based on the above fluorescence spectra of five systems at 25 °C and 35 °C, respectively. The results are shown in Table 1, and the corresponding linear regression plots are presented in Appendix A.

From Table 1, all the *K*_q_ values in five systems at 25 °C and 35 °C were remarkably greater than the maximum dynamic collision quenching constant (2 × 10^12^ L·mol^−1^·s^−1^) [29]; they deduced that the main fluorescence quenching mechanism of azalomycin F to LtaS was the static quenching, indicating a spontaneous process of azalomycin F binding LtaS. This was further confirmed since all the Δ*G* values were less than zero in the five tested systems. Differently, the *K*_sv_ values gradually decreased along with the increase in the incubation temperature for systems (A) to (C), while it slightly increased for systems (D) and (E). This indicated that a static and dynamic combined quenching mechanism [30], but mainly a static quenching one, should be responsible for the binding process of azalomycin F to LtaS, which is influenced by the competitive binding of azalomycin F to the DPPG of LtaS-embedded liposomes. However, this adverse effect from DPPG on the binding of azalomycin F to LtaS can be partly balanced by manganese ions; the *n*_H_ value of system (E) at 25 °C was less than that at 35 °C, while that at 25 °C was greater than that at 35 °C for system (D).

All the *K*_a_ values dropping into the range from 1 × 10^4^ to 1 × 10^6^ L·mol^−1^ indicated that azalomycin F can bind to LtaS with a moderate strength [31,32,33], and the number of binding sites ranged from 0.531 to 1.019, indicating that azalomycin F interacted with LtaS through one binding site. Simultaneously, all the *ΔG* values obviously less than zero also indicated a spontaneous process was generated for azalomycin F binding to LtaS [34]. Both the *ΔH* and *ΔS* values more than zero in system (A) indicated that the hydrophobic interactions played a main role for the binding process of azalomycin F to eLtaS, while the van der Waals forces and hydrogen bonds were mainly responsible for those in systems (B) and (C). This was likely due to the influence of the hydrophobic fatty acyl chain of DPPG, which was also supported by the fact that the hydrophobic interactions played a main role for the binding processes of azalomycin F to LtaS in systems (D) and (E) in which the fatty acyl chain of DPPG was unexposed to azalomcyin F and eLtaS. Although both the *ΔH* and *ΔS* values of system (E), which were more in line with the biochemical environment of LtaS, were more than zero, they were the smallest ones among those of systems (A), (D) and (E), and simultaneously the *ΔS* value of system (E) was close to zero. These further indicated that the hydrophobic interactions should play the main role for the binding process of azalomycin F to LtaS [28,34,35], which was also supported from the subsequent molecular docking.

Among the thirteen important amino acid residues reported in the LtaS active center, residues Trp354 and Tyr477, respectively, involve the binding of LtaS to the substrates and the stabilization of the growing LTA chain in the pocket [9]. To study the microenvironment information on the tyrosine or tryptophan residues of LtaS after azalomycin F bound to [36,37], the synchronous fluorescence spectra were determined for systems (A) to (E) with various azalomycin F concentrations, and the wavelength differences (Δ*λ*) between the emission and excitation wavelengths were kept as 15 and 60 nm, respectively. The results (Appendix A) showed that all the maximum emission wavelength (*λ*_max_) values were red-shifted along with the increase in azalomycin F concentration. This indicated that the polarity around the tryptophan and tyrosine residues of LtaS increased due to the interaction of azalomycin F with eLtaS or LtaS [38,39,40]. Considering the evidence that all the maximum fluorescence intensity values of these systems decreased along with the increase in azalomycin F concentration, it was further confirmed that there were interactions between azalomycin F and the tryptophan and tyrosine residues of LtaS during the binding process, which was also verified by the following experiments on mass spectroscopy analyses and molecule docking. However, the red shifts (about 3.8 to 5.6 nm) of *λ*_max_ for all systems when the Δ*λ* value was kept as 60 nm were greater than those (about 1.8 to 2.2 nm) of *λ*_max_ when it was kept as 15 nm. This showed that the influence of azalomycin F on the tryptophan residues was greater than that on the tyrosine ones.

### 2.5. Binding Site of Azalomycin F to LtaS

To explore the details of azalomycin F interacting with LtaS, the competitive binding of azalomycin F to various amino acids, including at the important sites of the active centers of LtaS, was analyzed by electrospray ionization (ESI) mass spectrometry (MS). The results (Appendix A) showed that three peaks with higher intensity at the mass-to-charge ratios (*m/z*) of 1066.6519, 1080.6668 and 1094.6844 corresponded to three [M-H] negative ions, respectively, from azalomycin F_3a_, F_4a_ and F_5a_. From the *m/z* range of 1100 to 1300 (Appendix A), another six peaks with higher intensity were also observed at the *m/z* of 1199.7171, 1213.7074, 1227.7144, 1235.7227, 1254.7797 and 1284.7588, corresponding to six complex ions [M-H] of azalomycin F_4a_, respectively, with aspartic acid (Asp), glutamic acid (Glu), tryptophan (Trp), threonine (Thr), histidine (His) and arginine (Arg). It indicated that azalomycin F had the potency to interact with many residues of LtaS active centers, especially with Asp, Glu and Trp ones, according to their peak intensities. Moreover, azalomycin F could also interact with phenylalanine (Phe) or tyrosine (Tyr) residues, which was deduced from their complex ions [M-H] and [M] with lower densities (Appendix A). This also coincided with the influence of azalomycin F on the Trp residues being greater than that on the Tyr ones of LtaS, which resulted from the above synchronous fluorescence spectra. Further analyses for the mass data (Appendix A) indicated that demalonyl azalomycin F_4a_ could also interact with the above amino acids, especially those which azalomycin F_4a_ more strongly bound to, such as Glu, Asp, Trp and Thr. This indicated that the malonyl moiety of azalomycin F does unlikely bind to the important sites of the LtaS active centers, which was consistent with previous conclusions that removing the malonyl moiety would increase the antibacterial activities of azalomycin F to *S. aureus* [13,19]. As the MS analyses for azalomycin F binding various amino acids of the LtaS active center were performed without negative controls, the results needed to require more supports from synchronous fluorescence spectra and molecular docking. In addition, it would further improve the reliability of these results if the analyses for the interactions of azlomycin F with various amino acid mutants of LtaS were performed based on these results.

To further explore the site and position of azalomycin F binding to LtaS, a molecular docking was conducted, focusing on the active site of eLtaS. As we reported [19], the lactone rings of azalomycin F_3a_, F_4a_ and F_5a_ can tightly bind to the polar head of cell membrane phospholipids, while their guanidyl side chains target the LTA. This is very similar to the spatial structure of LtaS, embedding in the cell membrane and its extracellular domain extending to out of the membrane. Thereby, three azalomycin F components and their guanidyl side chains were, respectively, selected for the molecular docking to eLtaS. All results (Figure 6) indicated that the hydrophobic interactions play a main role for azalomycin F binding to eLtaS, which also coincided with those results of the Δ*H* and Δ*S* calculations from the fluorescence spectra of five systems (Table 1). Furthermore, the docking results were also summarized in Appendix A, which indicated that there were obvious interactions of azalomycin F with the residues His416, Trp354 and Lys299 of eLtaS: whether its three components or their guanidyl side chains. This was also supported by MS analyses which showed the strong bindings of azalomycin F_4a_ to histidine and tryptophan, and by the results from the synchronous fluorescence spectra. Appendix A indicates that azalomycin F_4a_ can interact with three residues His476, Tyr477 and Gly478, while azalomycins F_3a_ and F_5a_ interact with just with one of both residues His476 and Tyr477 besides Gly478. This likely gave a reasonable interpretation for a slight difference from the antimicrobial activities of these three components [13,41]. Moreover, azalomycin F can probably interact with three residues, Glu255, Ser256, Phe353 (close to Trp354), and/or Ser480 (close to His476 to Gly478), from Appendix A.

According to previous studies [42], there are eleven amino acid residues involving the active center of eLtaS, which include a catalytic residue Thr300 binding to the metal ions, three residues Glu255, Asp475 and His476 binding to the metal ions, a residue His416 binding to the substrates and protonating groups, six residues His347, Asp349, Phe353, Trp354, Arg356 and Leu384 binding to the substrates, and two residues Tyr477 and Lys299 stabilizing the growing LTA chain in the pocket [9]. Thereby, the docking results (Appendix A) suggested that the guanidyl side chain of azalomycin F can interact with the eLtaS active center to inhibit the LTA biosynthesis, which involved some important sites binding to the substrates and metal ions and those stabilizing the growth of the LTA chain. This was also supported by the antimicrobial structure–activity relationship, indicating that the guanidyl side chain is an essential moiety for the polyhydroxy macrolides to inhibit Gram-positive bacteria [13,19,43].

Whether the guanidyl side chain alone or of azalomycin F can well dock into the active pocket and interact with various residues of the eLtaS active center, although their binding sites were slightly different. Simultaneously, most interactions occurred on the guanidyl side chain of azalomycin F, and only one to three hydrophobic ones occurred on its malonyl moiety. These together with the above MS analyses provided a reasonable interpretation that the antimicrobial activities of azalomycin F were enhanced after its malonyl moiety was removed [13,44], and some changes in the lactone rings of guanidyl-containing polyhydroxy macrolides have little influence on their antibacterial activities [13], such as the ring size and hydrogenation of some olefinic bonds.

Our previous works confirmed that the synergistic effect of azalomycin F binding to the polar head of the phospholipid through its lactone ring and targeting lipoteichoic acid through its guanidyl side chain was responsible for its anti-MRSA activity [19]. Here, this study suggested that azalomycin F can accelerate LTA release through a synergism of the membrane damage and the electrostatic interactions between its guanidyl group and the phosphate diester group of LTA, which together with the membrane damage would lead to the disruption of the cell envelope. Simultaneously, azalomycin F can also inhibit LTA biosynthesis through the binding of its guanidyl side chain to the amino acid residues of the active center of LtaS. Thereby, the synergistic action on the cell envelope disruption and the LTA synthesis inhibition should be responsible for the antimicrobial activities of azalomycin F, which would cause the leakage of cellular substance and eventually lead to the autolysis of bacterial cells. Accordingly, this would also lead to the feedback upregulation of LtaS and some related proteins.

LtaS was considered as an attractive antimicrobial target. However, there are no other LtaS inhibitors reported to date besides compound 1771, dye Congo red and compound 4 [8,9,10]: especially none reported from natural products. As mentioned in the Introduction section, except for dye Congo red, compound 1771 and its derivatives [11], together with two compounds reported by Muscato, et al. [12], were proved to not be LtaS inhibitors. Thereby, azalomycin F can be considered as the first natural LtaS inhibitor, and this research can provide an important reference for the discovery of more potent LtaS inhibitors to treat *S. aureus* infection. It can also possibly be used as a probe to understand how inhibiting lipoteichoic acid biosynthesis affects the cell physiology of Gram-positive bacteria. Furthermore, there are approximately sixty guanidyl-containing polyhydroxy macrolides discovered from natural products [13], and so it is valuable for exploring the antimicrobial mechanism of guanidyl-containing polyhydroxy macrolides. It may be reasonable to deduce that these guanidyl-containing polyhydroxy macrolides were likely LtaS inhibitors as they have similar lactone rings and guanidyl side chains to azalomycin F.

## 3. Materials and Methods

### 3.1. Materials, Chemicals and Reagents

*S. aureus* ATCC 25923 was purchased from the American Type Culture Collection, Manassas, VA, USA. *Escherichia coli* (*E. coli*) BL21(DE3) and the vector pET-28a(+) were purchased from Beijing Tsingke Biotechnology Co., Ltd. (Beijing, China). Azalomycin F (purity, 98%) was isolated from the fermentation of *Streptomyces hygroscopicus* var. *azalomyceticus* according to our previous method [14], and its MIC against *S. aureus* ATCC 25923 was 4.0 μg/mL, testing with the broth microdilution method [19]. Before use, the stock solution (2048 μg/mL) of azalomycin F was prepared by dissolving in dimethyl sulfoxide (DMSO), and the DMSO concentrations in all test systems were kept at less than 0.78% and even less than 0.031% in various incubation systems used for the interactions of azalomycin F with eLtaS or the LtaS-embedded liposome. The enzyme-linked immunosorbent assay (ELISA) kit for LTA was manufactured by Shanghai Enzyme-linked Biotechnology Co., Ltd. (Shanghai, China), and the ELISA kit for diacylglycerol (DAG) was manufactured by Cloud-Clone Corp. (Wuhan, China). The Ni^2+^-NTA-Sepharose was obtained from QIAGEN (Hilden, Germany).

Media used for the culture of *S. aureus* were purchased from Haibo Biotechnology Co., Ltd. (Qingdao, China). Protein markers for gel analyses were purchased from Thermo Fisher Scientific (Waltham, MA, USA), and most chemical reagents for the expressions, gel analyses and Western blot of eLtaS and LtaS were purchased from Sangon Biotechnology (Shanghai) Co., Ltd. (Shanghai, China) and Proteintech Group, Inc (Wuhan, China). All other chemical reagents were purchased from Sigma-Aldrich (St. Louis, MO, USA), Sinopharm Chemical Reagent Co., Ltd. (Shanghai, China) or Xilong Scientific Co., Ltd. (Shantou, China).

### 3.2. Influences of Azalomycin F on the LTA Release of S. aureus

Pure *S. aureus* colonies were inoculated into brain–heart infusion (BHI) medium and cultured at 37 °C for 24 h on a rotary shaker (160 rpm). Next, a 1:100 dilution of this culture was prepared using fresh trypticase soy broth (TSB) and incubated at 37 °C until the OD_590_ value of about 0.8. After the bacterial suspension was centrifuged at 12,000 rpm for 10 min, the pellet was diluted using TSB to obtain a bacterial suspension with the OD_590_ value of 0.20. To this suspension, the bacterial growth was interfered with different concentrations of azalomycin F (0, 1, 2, 4 and 8 μg/mL), and 1 mL of sample was taken with an interval of 30 min. The sample was centrifuged at 4000 rpm for 15 min to obtain the supernatant, and then the LTA contents in this supernatant were determined using the ELISA kit for LTA according to the described procedure. The experiment was performed three times.

### 3.3. Transmission Electron Microscope (TEM)

A 1:100 dilution of overnight culture with fresh Muller–Hinton broth (MHB) was incubated at 37 °C until the exponential phase. After the bacterial suspension was centrifuged at 4000 rpm for 10 min, the pellet was resuspended into the sterile phosphate-buffered saline (PBS) solution containing different concentrations of azalomycin F (0, 4 and 8 μg/mL), and it stayed at 37 °C for 6 h on a rotary bed. The suspension was centrifuged at 4000 rpm for 10 min and washed with PBS solution. Then, the cells were solidified overnight with a 2.5% glutaraldehyde solution at 4 °C, fixed in 1% osmic acid for 2 h, and gradually dehydrated in a series of ethanol solutions [43]. The dehydrated samples were successively dealt with the solutions of embedding agent mixed with acetone at a ratio of 1:1 and 3:1, for 1 to 3 h, and then with an embedding agent at 70 °C overnight. For TEM observation, the embedded samples were cut into 70 to 90 nm slices on a Leica EM UC7 ultramicrotome (Leica Microsystems, Wetzlar, Germany), and then the slices were successively stained with lead citrate solution and 50% ethanol saturated solution of uranyl acetate for 10 min.

### 3.4. Proteome

A 1:100 dilution of overnight culture with fresh TSB was incubated at 37 °C until the OD_590_ value of 0.30, and three drug groups containing 8.0 μg/mL azalomycin F and three blank ones containing the same concentration of DMSO in the drug groups were prepared. After being incubated at 37 °C for 4 h, bacterial cells were obtained by centrifugating at 4000 rpm for 10 min, repeatedly washing three times with PBS, and ultrasonication three times in a lysis buffer (8 M urea, 1% Protease Inhibitor Cocktail). The supernatant was collected after the remaining debris was removed by centrifugation (12,000 rpm) at 4 °C for 10 min. Then, the protein concentration was determined with a BCA protein assay kit (Thermo Fisher Scientific, Waltham, MA, USA) according to the manufacturer’s instructions. After trypsin digestion, the peptides were desalted by a Strata X C18 SPE column (Phenomenex, Torrance, CA, USA) and vacuum-dried. Then, the peptides were reconstituted in 0.5 M TEAB and labeled according to the manufacturer’s protocol for a TMT kit (Thermo Fisher Scientific, Waltham, MA, USA). Subsequently, the peptide mixtures were incubated at room temperature for 2 h, pooled, desalted, and dried by lyophilization. Finally, the tryptic peptides were separated into 60 fractions by a high pH reverse-phase HPLC using a Thermo Betasil C18 column (5 μm particles, 4.6 mm i.d., 250-mm length) with a gradient of 8% to 32% acetonitrile (pH 9.0) in 60 min. After combination, 18 fractions were obtained and dried by lyophilization.

The peptides from tryptic hydrolysis were dissolved in 0.1% (*v/v*) formic acid and separated on an EASY-nLC 1000 UPLC system (Thermo Fisher Scientific, San Jose, CA, USA) with a home-made reversed phase analytical column (15-cm length, 75 μm i.d.). The 2% acetonitrile mixture contained 0.1% (*v/v*) formic acid and a 90% acetonitrile contained 0.1% (*v/v*) formic acid, which were used as mobile phases A and B, respectively. The gradient elution consisted of an increase from 6% to 22% of solvent B in 42 min, which was followed by an increase from 22% to 30% in 12 min, 30% to 80% in 3 min, and then holding at 80% for the last 3 min. The analysis process was achieved at a constant flow rate of 0.5 μL/min. The separated peptides were subjected to an NSI source followed by MS/MS analysis in a Q Exactive^TM^ Plus (Thermo Fisher Scientific, San Jose, CA, USA) coupled online to the UPLC. The electrospray voltage applied was 2.4 kV. The full scan of *m/z* ranged from 350 to 1550, and intact peptides were detected in the Orbitrap at a resolution of 60,000. The peptides were then selected for MS/MS analysis using the NCE setting as 28, and the fragments were detected in the Orbitrap at a resolution of 15,000. A data-dependent procedure alternated between one MS scan followed by 20 MS/MS scans with a 30 s dynamic exclusion. Automatic gain control (AGC) was set at 5e^4^. The fixed first *m/z* was set as 100.

The obtained MS/MS data were processed using the Maxquant search engine (v.1.5.2.8). Tandem mass spectra were searched against the *Staphylococcus aureus* database concatenated with reverse decoy database. Trypsin/P was specified as the cleavage enzyme, allowing up to two missing cleavages. The mass tolerances for precursor ions were set as 20 ppm in the first search and 5 ppm in the main search, and 0.02 Da was set for those of the fragment ions. Carbamidomethyl on *Cys* was specified as a fixed modification, while oxidation on *Met*, acetylation modification on protein *N*-terminal, and deamidation were specified as variable modifications. For the quantification method, TMT 6-plex was selected. The false discovery rate (FDR) was set as 1%, and the minimum score for modified peptides was set as more than 40. For subsequent bioinformatics analysis, after the mass errors and lengths of all the identified peptides were checked and fit the corresponding requirements, differentially expressed proteins were divided into two categories. A protein with a 1.2 (or more) quantitative ratio of azalomycin F group to the blank one was considered as the upregulated one, while that less than 1/1.2 was considered as downregulated one. The Gene Ontology (GO) annotation of the proteome was performed using the UniProt-GOA Database (http://www.ebi.ac.uk/GOA/), accessed on 5 March 2019. Proteins were categorized into biological processes, cellular components, and molecular functions according to the GO annotation. The KEGG was utilized to annotate pathways. For functional enrichment analysis, a two-tailed Fisher’s exact test was used to assess the enrichment or depletion of specific annotation terms among members of the resulting protein clusters. Any terms with adjusted *p* values below 0.05 or 0.01 in any of the clusters were considered as having a significant or remarkably significant difference.

### 3.5. Targeted Protein Quantification

Referred to the proteomic results, six related proteins, lipoteichoic acid synthase (Q2G093), signal peptidase I (Q2FZT7), teichoic acids export ATP-binding protein TagH (Q2G2L1), lipoprotein (Q2G0V0), UDP-N-acetylmuramoyl-tripeptide-D-alanyl-D-alanine ligase (Q2FWH4), and D-alanine-D-alanine ligase (Q2FWH3), were selected for further verifiable quantification for three samples from the azalomycin F group or the blank one, using the parallel reaction monitoring (PRM) technology. According to the similar procedure to proteome analysis, the peptides from tryptic hydrolysis were dissolved in 0.1% (*v/v*) formic acid and separated on an EASY-nLC 1000 UPLC system with a reversed phase analytical column. The gradient elution consisted of an increase from 7% to 25% solvent B in 40 min, 22% to 35% in 12 min, 35% to 80% in 4 min, and then holding at 80% for the last 4 min. The analysis process was achieved at a constant flow rate of 0.4 μL/min. The separated peptides were subjected to an NSI source followed by MS/MS analysis in a Q ExactiveTM Plus (Thermo Fisher Scientific, San Jose, CA, USA). The electrospray voltage applied was 2.0 kV. The full scan of *m*/*z* ranged from 400 to 1000, and intact peptides were detected in the Orbitrap at a resolution of 70,000. The peptides were then selected for MS/MS analysis using an NCE setting as 27, and the fragments were detected in the Orbitrap at a resolution of 17,500. The data-dependent procedure alternated between one MS scan followed by 20 MS/MS scans. The AGC was set at 3e^6^ for full MS and 1e^5^ for MS/MS, respectively. The maximum IT was set at 50 ms for full MS and 180 ms for MS/MS. The isolation window for MS/MS was set at 1.6 *m/z*.

The resulting MS data were processed using Skyline (v.3.6). Peptide parameters were set as trypsin (KR|P) for the enzyme, zero for the max missed cleavage, 7 to 25 residues for the peptide length, and alkylation on Cys for fixed modification. Transition parameters were set as 2 and 3 for precursor charges, 1 for ion charges, and *b* and *y* for ion types. The product ions were set from ion 3 to the last ion. The mass tolerance of ion match was set as 0.02 Da.

### 3.6. Influence of Azalomycin F on the LTA Synthesis from LtaS

The eLtaS was expressed in *Escherichia coli* BL21(DE3)/pET28a(+)-eLtaS. Briefly, *E. coli* BL21(DE3)/pET28a(+)-eLtaS was cultured onto a Luria–Bertani (LB) agar plate at 37 °C, and then pure colonies were inoculated into LB broth that contained kanamycin (50 μg/mL) on a rotary shaker (180 rpm) for 12 h at 37 °C. Next, a 1:100 dilution with fresh LB broth containing kanamycin (50 μg/mL) was cultured at 37 °C until the OD_600_ value of 0.6 to 0.8. After cooling for 15 min, isopropyl *β*-D-1-thiogalactopyranoside (IPTG) (Sangon Biotechnology (Shanghai) Co., Ltd., Shanghai, China) with the final concentration of 1 mM was added into the culture. Finally, the expression was induced on a rotary shaker (180 rpm) for 12 h at 18 °C, and the bacterial cells were collected by centrifugation at 4000 rpm for 15 min and stored at −20 °C. When in use, the bacterial cells in a lysis buffer (50 mM Tris-Cl, 200 mM NaCl, 5% glycerol, 0.25% *n*-dodecyl-*β*-D-maltopyranoside, pH 8.0) were crushed and centrifugated by 12,000 rpm for 10 min at 4 °C. Then, the LtaS in the supernatant was purified according to the similar methods described previously [25,42]. The concentrations of purified eLtaS were determined using a modified BCA protein assay kit, and the purity of eLtaS was determined by SDS–PAGE and Western blot. Similarly, the LtaS was expressed in *E. coli* BL21(DE3)/pET28a(+)-LtaS and purified using the Ni^2+^-NTA-Sepharose affinity column. The purity of LtaS was also determined using SDS–PAGE and Western blot methods, while the concentrations of purified LtaS were determined by the ImageJ method using the BCA protein for preparing a series of standard solutions.

Referring to previous papers [19], large unilamellar vesicles (LUVs) without LtaS were prepared. Briefly, 15 mL of CHCl_3_:CH_3_OH (2:1, *v/v*) was added to a centrifuge tube containing 4 mL wet *S. aureus* cells in an exponential phase. After the mixture was vigorously vortexed for 30 min, 5 mL of CHCl_3_ was added with a subsequent vortex of 15 min, and then another 5 mL of 10 mM Tris-HCl (pH 3.0) was added followed by a vortex of 15 min. The mixture was centrifuged at 4000 rpm for 10 min, and the organic phase was collected and evaporated on a rotavapor system. Subsequently, the dried films containing the cell membrane phospholipids of *S. aureus* were obtained after being dried under nitrogen flow. The films were hydrated with 1 mL of 10 mM Tris-HCl (pH 7.4) at 50 °C to obtain multilamellar vesicles (MLVs) suspension, and then LUVs were prepared by extrusion of the MLVs suspension through a polycarbonate filter with 100 nm of diameter in a 610020-1EA Mini-Extruder (Avanti Polar Lipids Inc., Alabaster, USA), which was kept at 60 °C. The apparent average particle sizes and surface charges of LUVs were determined in triplicate by a Zetasizer Nano S90 (Malvern Instruments Ltd., Malvern, Worcestershire, UK) with a 90° of scattering angle at 25 °C, and the test concentrations of all vesicles were 0.5 mg/mL.

The LUVs were diluted with twice the volume of liposome diluent (25 mM Tris (pH 8.0), 200 mM NaCl, 0.4% *n*-dodecyl *β*-D-maltopyranoside, pH 7.5). After being pre-cooled in an ice bath for 10 min, the LtaS was added to make a final concentration of 1 μM. The solution was shaken for 1 h at 4 °C and then mixed with an equal volume of Bio-beads SM2 (Bio-Rad Laboratories, Inc., Hercules, CA, USA) equilibrated in 25 mM Tris (pH 8.0), 200 mM NaCl and 100 μM EDTA, shaking at 4 °C for 4 h and replacing fresh Bio-Beads SM-2 for 4 repetitions to ensure the complete removal of surfactants. For removing the unembedded proteins, the mixture was centrifugated at 10,000 rpm at 4 °C for 10 min, and then the LtaS-embedded liposomes in the supernatant were obtained and stored at 4 °C with a shelf life of 25 to 30 d.

According to the method reported by Karatsa-Dodgson [25], an ELISA assay kit was used to detect the product diacylglycerol (DAG) from enzymatic reaction with three repeats for verifying the eLtaS activity and exploring the influence of azalomycin F on the LTA synthesis from eLtaS. Briefly, 25 μg L-*α*-phosphatidylacyl-DL-glycerol (sodium salt) (PG) (Sigma-Aldrich, St. Louis, MO, USA) was added into a sodium succinate buffer (10 mM, pH = 6.5) and dissolved by 1 min of ultrasonication. To this solution, 1 M MnCl_2_ was added to give a final concentration of 10 mM, and then 100 μg eLtaS was added into for initiating the enzyme reactions. The reaction systems containing various concentrations of azalomycin F (0, 10, 20, 40, 80, 160, 320 and 640 μM) were incubated at 37 °C for 3 h, and then the ELISA kit was used to detect the product DAG. The incubation systems without eLtaS were used as blank groups. Similarly, the enzyme activities of LtaS-embedded liposomes in the presence of different concentrations of azalomycin F (0, 10, 20, 40, 80 and 160 μM) were evaluated, and the incubation system containing the liposomes without chimeric proteins LtaS was used for the blank groups.

### 3.7. Interaction of Azalomycin F with the eLtaS or LtaS of S. aureus

#### 3.7.1. Fluorescence Spectroscopy

An FL970 fluorescence spectrophotometer (Techcomp Instrument Ltd., Shanghai, China), setting the slit width as 2.5 nm and the sampling interval as 0.2 nm, was employed for recording the fluorescence spectra of five incubation systems: (A) eLtaS/azalomycin F, (B) eLtaS/DPPG/azalomycin F, (C) eLtaS/DPPG/MnCl_2_/azalomycin F, (D) LtaS-embedded liposome/azalomycin F and (E) LtaS-embedded liposome/MnCl_2_/azalomycin F. The experiment was performed three times. All components in these systems were added into the sodium succinate buffer (10 mM, pH = 6.5), and the concentrations of eLtaS, the LtaS-embedded liposome, DPPG (Sigma-Aldrich, St. Louis, USA) and MnCl_2_ were 5, 5, 5 and 10 mM, respectively. Nine interfered concentrations—0 (blank group), 5, 10, 15, 20, 25, 30, 35 and 40 μM—were set for azalomycin F in systems (A) to (C), while ten interfered concentrations—0 (blank group), 2, 4, 8, 10, 20, 40, 60, 80 and 100 μM—were set for azalomycin F in systems (D) and (E). After being incubated for 5 min in a water bath at 25 °C and 30 °C for each system, the fluorescence spectra from 290 to 400 nm or to 450 nm were recorded at the corresponding incubation temperature with the excitation wavelength (*λ*_ex_) at 280 nm. From the spectra, the maximum emission wavelengths and fluorescence intensities were recorded, and the quenching constants, the thermodynamic parameters, and the Hill’s coefficients (*n*_H_) at different temperatures were calculated according to the methods and formula described in the Appendix A (1. Experiments Details) [28,31,32,33,35,45,46] for exploring the quenching mechanism and the binding forces, understanding the binding constant and the number of binding sites, of azalomycin F interacting with eLtaS or LtaS. The fluorescence intensity also needed to be corrected to decrease the inner filter effect according to the following formula [47,48]:*F*_cor_ = *F*_obs_ × *e*^(*A*ex + *A*em)/2^(1)
where *F*_cor_ and *F*_obs_ represent the corrected and observed fluorescence intensities, respectively; and *A*_ex_ and *A*_em_ represent the absorptions of sample solutions at the excitation and the emission wavelengths, respectively.

#### 3.7.2. Synchronous Fluorescence Spectra

The synchronous fluorescence spectra of the sample solutions for the emission spectral determinations were further measured at the temperature of 25 °C with the *λ*_ex_ from 250 to 350 nm or from 230 to 390 nm, respectively, setting the difference values (Δ*λ*) between the emission and the excitation wavelengths fixed at 15 nm and 60 nm. From the spectra, the maximum emission wavelengths and fluorescence intensities were recorded. The experiment was performed three times.

### 3.8. Mass Spectrometry

Eight amino acids including tyrosine, tryptophan, threonine, glutamic acid, aspartic acid, histidine, phenylalanine, and arginine (Sinopharm Chemical Reagent Co., Ltd., Shanghai, China) were added into a 30% methanol aqueous solution of azalomycin F (100 μM) to obtain a mixed solution by vortexing, according to the molar concentration ratio of azalomycin F to each amino acid of 1:8. Simultaneously, the 30% methanol aqueous solution that only contained 100 μM azalomycin F was prepared as a blank control for background subtraction. After being incubated in a water bath at 37 °C for 30 min, these solutions were analyzed by a TripTOF 5600+ mass spectrometry (AB Sciex, Framingham, MA, USA) with an electrospray ion resource, and the detailed parameters are shown in Appendix A for negative ionization modes.

### 3.9. Molecular Docking

Using software Autodock Tools 1.5.7 and Autodock Vina 1.1.2, the molecular dockings between azalomycins F_3a_, F_4a_, F_5a_ or their guanidyl side chains and eLtaS were performed, respectively. Briefly, the crystal structures of LtaS (PDB ID: 2W5Q) were download from the PDB online platform (https://www.rcsb.org/ accessed on 5 March 2019), and the ions and water molecules were deleted by Pymol 2.5 [49]. The three-dimensional (3D) structures of azalomycins F_3a_ (code: 73425468), F_4a_ (code: 76963406) or F_5a_ (code: 73425500) were downloaded from PubChem (https://pubchem.ncbi.nlm.nih.gov/ accessed on 5 March 2019), while the 2D structures for the guanidyl side chains of azalomycins F_3a_, F_4a_ and F_5a_ were obtained by ChemBioDraw Ultra 14.0. All the structures of the ligands were imported into software ChemBio3D Ultra 14.0 to generate the 3D structure with minimal energy states. Referring to the related reports [8,9,10], the active pocket of eLtaS was predicted by the DrugRep tool [50]. Autodock Tools 1.5.7 was used for adding charge and polar hydrogen atoms and the format conversion [51]. The grid size was set to 60 Å × 70 Å × 60 Å along the *X* (−19.939 Å), *Y* (3.504 Å) and *Z* (−37.427 Å) axes with 0.375 Å. Autodock Vina 1.1.2 was applied to calculate the possible conformation between azalomycin F and eLtaS. The conformation with the lowest binding free energy was selected and imported into Pymol 2.5 and Ligplot 2.2 for further processing, analysis, and generation images showing the 3D structure of the docking model [52].

### 3.10. Statistical Analysis

All statistical analyses were performed using the Excel program of Microsoft Office 2016, and the results were expressed as mean ± SD from at least three experiments. Before we employed parametric analysis, the normality and homoscedasticity tests for all statistical data were performed using the software of the Data Processing System (DPS, Zhejiang University, Hangzhou, China). The *p* values for the comparison of two groups were calculated using a bilateral *t*-test. A *p* value less than 0.05 shows that the data difference between two groups is significant, and one less than 0.01 indicates that the data difference between two groups is very significant. The correlation analyses and regression equations, together with their correlation coefficients (*r*), were performed using scatter plot and curve-fitting tools.

## 4. Conclusions

In summary, this study concluded azalomycin F has multiple antibacterial mechanisms against *S. aureus*. It kills *S. aureus* not only by the inhibition on the LTA biosynthesis through the interactions of its guanidyl side chain with some important residues of LtaS active center but also by the disruption to the cell envelope through accelerating the LTA release, damaging the cell membrane, and electrostatically interacting with LTA. Simultaneously, these antibacterial mechanisms exhibit a synergistic inhibition effect on *S. aureus* cells, which would eventually cause the autolysis of bacterial cells and the feedback upregulation of LtaS and some related proteins. Since azalomycin F is the first natural LtaS inhibitor reported, this research can also provide an important reference for the discovery of more potent LtaS inhibitors to treat *S. aureus* infection. It can also possibly be used as a probe to understand how inhibiting lipoteichoic acid biosynthesis affects the cell physiology of Gram-positive bacteria.

## Figures and Tables

**Figure 1 molecules-29-00856-f001:**
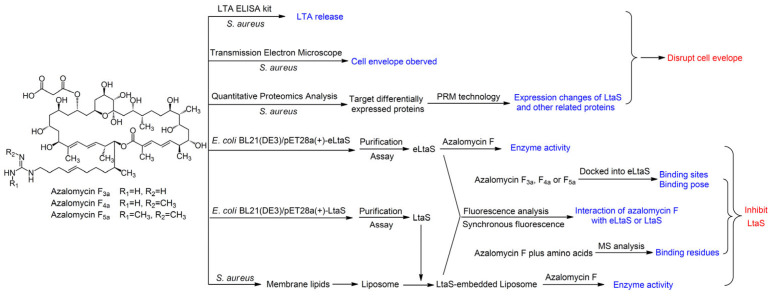
The research route for azalomycin F targeting the LTA of *Staphylococcus aureus*.

**Figure 2 molecules-29-00856-f002:**
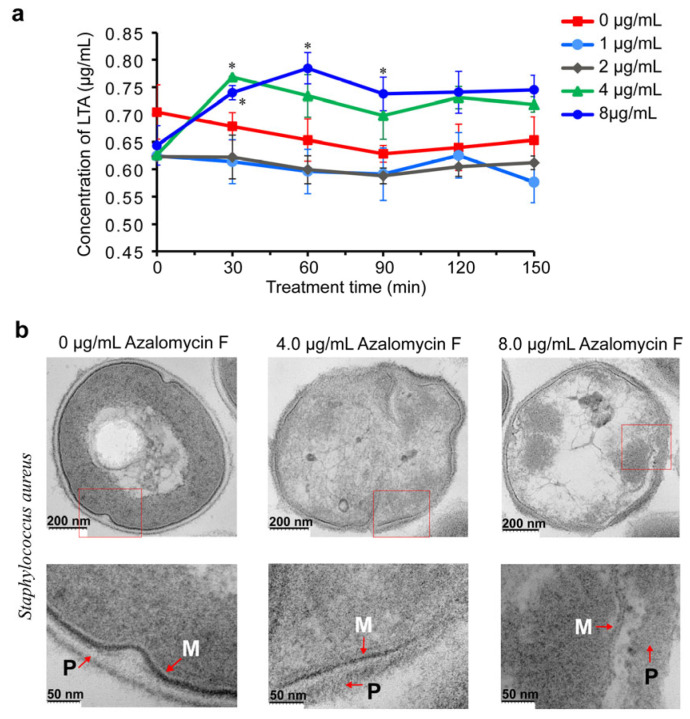
Azalomycin F accelerating the release of cellular LTA. (**a**) The influences of azalomycin F on the LTA release of *S. aureus* (*n* = 3); symbols * showed significant differences at *p* < 0.05 level, compared with the blank groups (0 μg/mL) at the same timepoints; the LTA concentration values of longitudinal coordinates were normalized by the OD_590_ values of bacterial suspensions. (**b**) Thin-section transmission electron micrographs of *S. aureus* ATCC 25923 after being treated with 4.0 or 8.0 μg/mL azalomycin F at 37 °C for 6 h. Upper, electron micrographs of *S. aureus*; Below, enlarged image of rectangles in the upper micrographs. M, plasma membrane; P, peptidoglycan layer. Scale bars are indicated at the bottom left of each micrograph.

**Figure 3 molecules-29-00856-f003:**
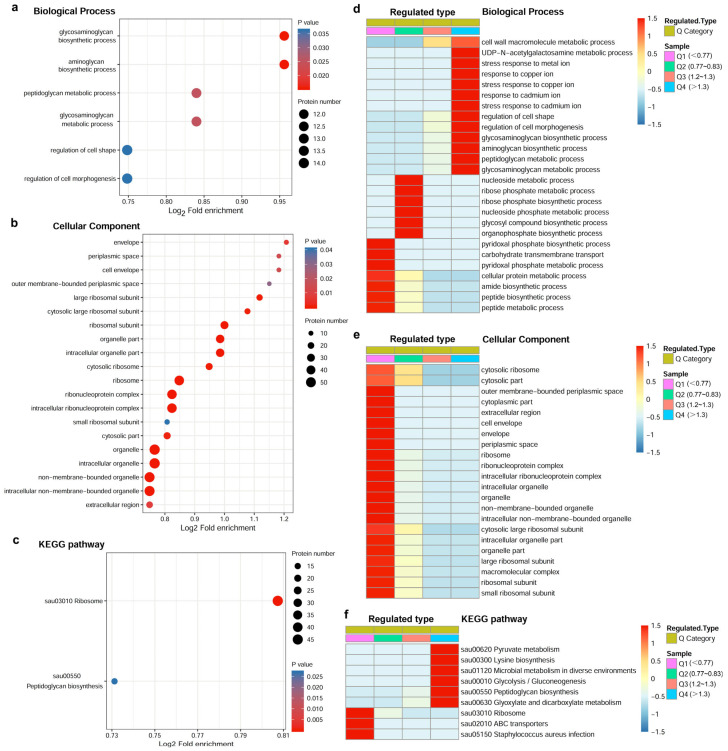
Bioinformatics analysis for the differentially expressed proteins of *S. aureus* treated with and without azalomycin F (*n* = 3). (**a**,**b**) Bubble plots for the Gene Ontology (GO) functional enrichment of differentially expressed proteins in biological process and cellular component, respectively. (**c**) Bubble pot for the Kyoto Encyclopedia of Genes and Genomes (KEGG) pathway enrichment of differentially expressed proteins. (**d**,**e**) Heatmaps of cluster analyses for the GO functional enrichment of differentially expressed proteins in biological process, cellular component and molecular function. (**f**) Heatmaps of cluster analyses for the KEGG pathway enrichment of differentially expressed proteins.

**Figure 4 molecules-29-00856-f004:**
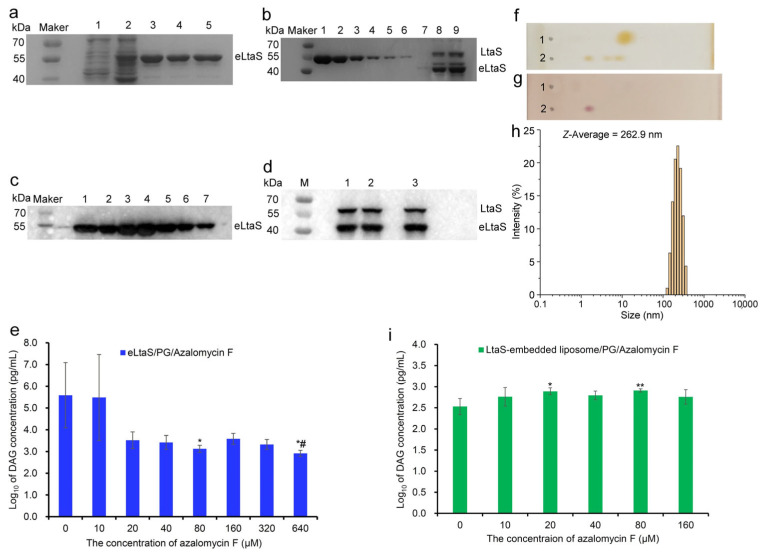
Preparations and analyses of eLtaS and LtaS-embedded liposomes, and the influence of azalomycin F on both enzyme activities. (**a**): SDS–PAGE analyses for eLtaS expressed in *E. coli* BL21(DE3)/pET28a(+)-eLtaS; lane 1, expressed without isopropyl-*β*-D-thiogalactopyranoside (IPTG) induction; lane 2, expressed with IPTG induction; lanes 3 to 5 were purified eLtaS proteins. (**b**): SDS-PAGE analyses for LtaS expressed from *E. coli* BL21(DE3)/pET28a(+)-LtaS; lanes 1 to 6 were BCA proteins with different concentrations respectively at 400, 200, 100, 50, 25 and 12.5 μg/mL; lanes 7 and 8 were, respectively, the last tube of elution buffer and the collected elution buffer from the Ni^2+^-NTA-Sepharose affinity column; lane 9 was LtaS solution concentrated by an ultrafiltration centrifuge tube from the collected elution buffer. (**c**,**d**): Western blot identifications, respectively, for eLtaS and LtaS, and various numbers represent the purified proteins. (**e**): Influence of azalomycin F on the LTA synthesis from eLtaS (*n* = 3), * and ^#^ indicated significant differences at *p* < 0.05 levels, respectively, compared with blank groups (0 μM) and 80 μM azalomycin F groups; (**f**,**g**): TLC analyses for the cell membrane phospholipids extracted from *S. aureus*, respectively, colored with iodine vapor and 0.5% ninhydrin/acetone solution. (**h**): The size distribution of LtaS-embedded liposomes. (**i**): Influence of azalomycin F on the LTA synthesis from LtaS-embedded liposomes (*n* = 3), * and ** indicated significant differences, respectively, at *p* < 0.05 and 0.01 levels compared with blank groups (0 μM).

**Figure 5 molecules-29-00856-f005:**
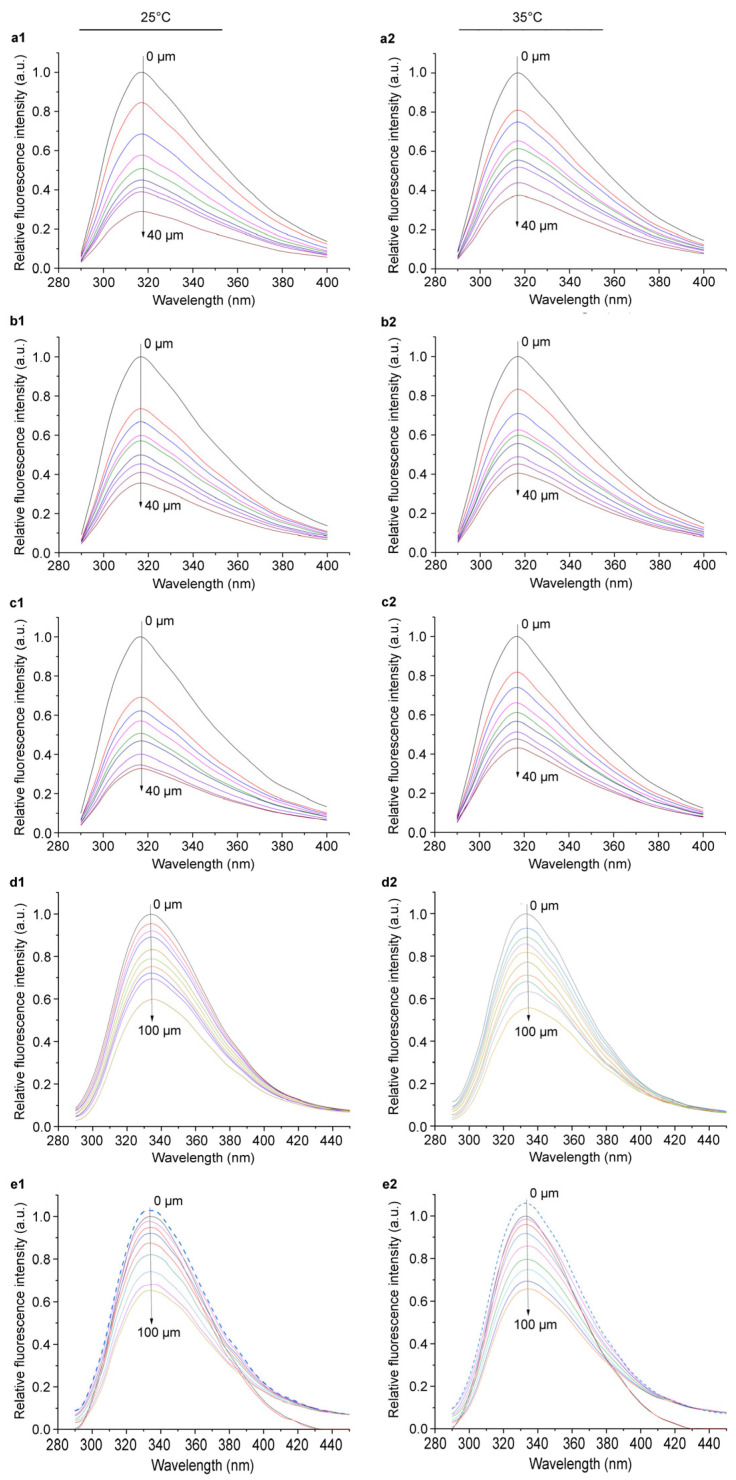
Fluorescence emission spectra of eLtas and LtaS quenched by azalomycin F. The spectra (**a1**–**e1**) of the left column corresponded to systems (A) to (E) at 25 °C, and (**a2**–**e2**) of the right column corresponded to those at 35 °C, respectively. The concentrations of azalomycin F were 0, 5, 10, 15, 20, 25, 30, 35 and 40 μM for systems (A) to (C), while they were 0, 2, 4, 8, 10, 20, 40, 60, 80 and 100 μM for systems (D) and (E). All those of the eLtaS and LtaS-embedded liposome were 5 μM.

**Figure 6 molecules-29-00856-f006:**
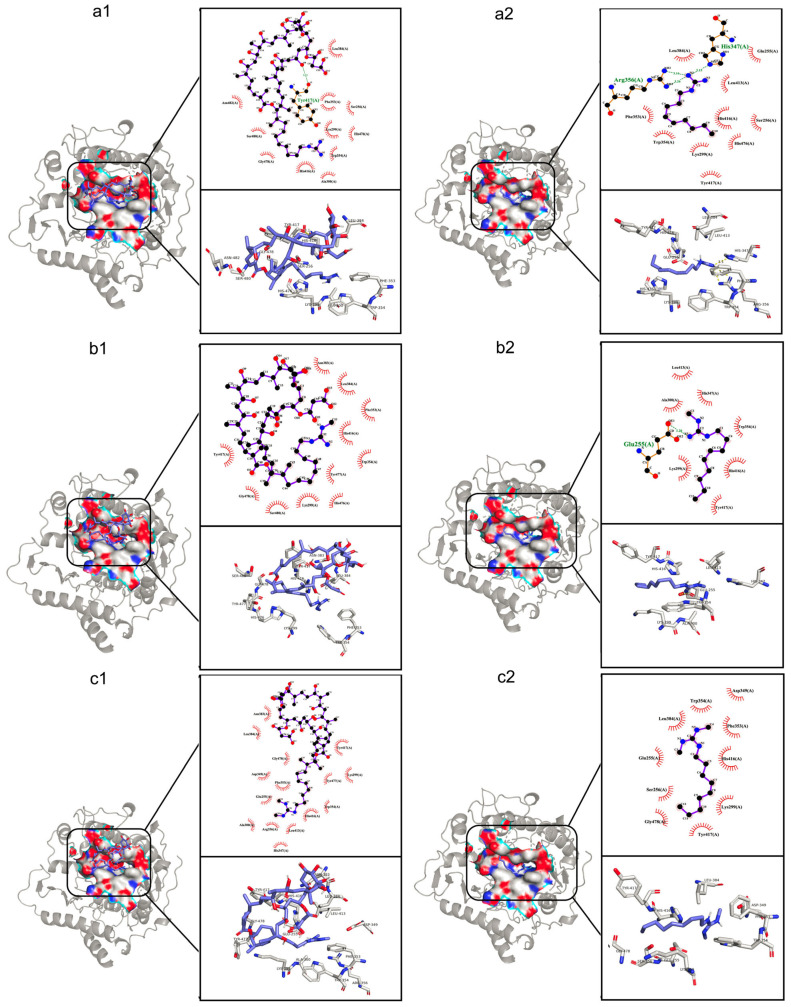
Molecular docking of azalomycin F and its guanidyl side chain alone into eLtaS. (**a1**,**b1**,**c1)** in the left panel show the 3D structures for azalomycins F_3a_, F_4a_ and F_5a_ docking into eLtaS, respectively; and (**a2**,**b2**,**c2**) in the right panel present the 3D structures for the guanidyl side chains alone of azalomycins F_3a_, F_4a_ and F_5a_ docking into eLtaS, respectively.

**Table 1 molecules-29-00856-t001:** Various parameters calculated from the fluorescence spectra of five systems at 25 °C and 35 °C (*n* = 3). ^a^

System ^b^	Temperature (°C)	*K*_sv_ (×10^4^ L·mol^−1^)	*K*_q_ (×10^12^ L·mol^−1^·s^−1^)	*K*_a_ (×10^4^ L·mol^−1^)	*n*	Δ*H* (KJ·mol^−1^)	Δ*S* (KJ·mol^−1^·k^−1^)	Δ*G* (KJ·mol^−1^)	*n* _H_
(A)	25	5.358	5.358	4.660	1.019	31.123	0.114	−2.803	/^c^
	35	3.751	3.751	4.543	0.918	31.123	0.114	−3.941	/
(B)	25	4.091	4.091	9.465	0.753	−48.796	−0.145	−5.568	/
	35	3.474	3.474	4.993	0.898	−48.796	−0.145	−4.118	/
(C)	25	4.881	4.881	11.495	0.748	−58.010	−0.174	−6.050	/
	35	3.100	3.100	5.375	0.842	−58.010	−0.174	−4.306	/
(D)	25	0.663	0.663	4.385	0.558	23.418	0.091	−3.662	0.503
	35	0.816	0.816	5.960	0.531	23.418	0.091	−4.571	0.438
(E)	25	0.531	0.531	1.366	0.773	11.347	0.041	−0.772	0.503
	35	0.546	0.546	1.585	0.755	11.347	0.041	−1.179	0.563

^a^ *K*_sv_, the Stern–Volmer quenching constant; *K*_q_, the quenching constant; *K*_a_, the binding constant; *n*, the number of binding sites; Δ*H*, enthalpy change; Δ*S*, entropy change; Δ*G*, Gibbs energy change; *n*_H_, Hill’s coefficient; and all the correlation coefficients for *K*_sv_, *K*_q_, *K*_a_, *n*, and *n*_H_ ranged from 0.964 to 0.998. All data were determined in triple, and the SD values are not shown. ^b^ Systems (A), eLtaS/azalomycin F; (B), eLtaS/DPPG/azalomycin F; (C), eLtaS/DPPG/MnCl_2_/azalomycin F; (D), LtaS-embedded liposome/azalomycin F; and (E), LtaS-embedded liposome/MnCl_2_/azalomycin F. ^c^ Uncalculated.

## Data Availability

No new data were created or analyzed in this study. Data sharing is not applicable to this article.

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
