# Peer review of "A Novel Antimicrobial Mechanism of Azalomycin F Acting on Lipoteichoic Acid Synthase and Cell Envelope"

_molecules, 2024, doi:10.3390/molecules29040856_

Round 1
Reviewer 1 Report
Comments and Suggestions for Authors
In this manuscript, the authors investigate the antimicrobial activity mechanism of Azalomycin F on S. aureus. While there is convincing evidence to support the interaction between Azalomycin F and LtaS, its relationship with LTA release remains unclear. I have the following comments on this manuscript:
1) Azalomycin F affects various processes in Gram-positive bacteria, suggesting that it exhibits a more general toxic mechanism. In this manuscript, the authors claim that the antimicrobial activity against S. aureus occurs by inhibiting the activity of LtaS. The authors should clearly mention in the abstract and conclusion sections whether the antimicrobial activity of Azalomycin F against S. aureus occurred due to the LtaS interaction or if it is one of the several mechanisms by which Azalomycin F exhibits antimicrobial activity.
2) Lines 98–104 and 127–130, Figure 2a: Azalomycin F did not demonstrate a concentration-dependent release of LTA; the amounts of LTA liberated at 4 and 8 µg/mL are similar in quantity. This might suggest that Azalomycin F does not have a direct relation to LTA release. Please provide comments on this or make appropriate changes to the manuscript.
Additionally, for lines 111–113, conducting a kinetic experiment using a biochemical assay for membrane integrity along with LTA release will shed more light on the relationship between LTA release and membrane damage, determining whether they are exclusive events or interdependent.
3) Lines 181–197: Please provide data and refer to the figure number for all expression level data. Additionally, if the authors verify the expression level data using alternative techniques such as RT-PCR, western blot, etc., it will enhance their hypothesis.
4) Lines 278–280, figure S1: The concentration of enzyme used for this assay is not mentioned.
5) Lines 319-321, figure 4e: Please put no enzyme control in this figure.
6) Section “2.5. Binding site of Azalomycin F to LtaS," Table S1: In this table, all tested amino acids show interaction with Azalomycin F; therefore, a negative control where Azalomycin F did not demonstrate any interaction with amino acids is required.
7) Materials and Methods section: Please mention the manufacturer of the ELISA kit for LTA and diacylglycerol.
Author Response
Dear Reviewer,
My co-authors and I are very grateful to you for your careful review, good comments, kind reminder and valuable suggestions. We have amended the manuscript according to the issues raised by you, and have pleasure to submit the revised version, together with the responses to your comments, for your consideration.
Many thanks for your kind attention!
Yours sincerely,
Ganjun Yuan
Here are our answers to your comments.
Comments and Suggestions for Authors
(1) In this manuscript, the authors investigate the antimicrobial activity mechanism of Azalomycin F on S. aureus. While there is convincing evidence to support the interaction between Azalomycin F and LtaS, its relationship with LTA release remains unclear. I have the following comments on this manuscript:
Response 1: Thank you for your careful review and enthusiastic help to improve our work! We had revised the manuscript according to your suggestion, and performed extensive modifications throughout the manuscript, for your consideration.
(2) Azalomycin F affects various processes in Gram-positive bacteria, suggesting that it exhibits a more general toxic mechanism. In this manuscript, the authors claim that the antimicrobial activity against S. aureus occurs by inhibiting the activity of LtaS. The authors should clearly mention in the abstract and conclusion sections whether the antimicrobial activity of azalomycin F against S. aureus occurred due to the LtaS interaction or if it is one of the several mechanisms by which Azalomycin F exhibits antimicrobial activity.
Response 2: Thank you very much for your help to improve our work! We had revised the abstract and conclusion sections, and expressed what you mentioned clear, for your consideration. For example, the last two sentence of the abstract were revised as follows:
“All these suggested that azalomycin F has multiple antibacterial mechanisms against S. aureus. It not only can inhibit LTA biosynthesis through the interactions of its guanidyl side chain with LtaS active center, but also can disrupt cell envelope through the synergistic effect of accelerating the LTA release, damaging the cell membrane, and electrostatically interacting with LTA. Simultaneously, these antibacterial mechanisms exhibit a synergistic inhibition effect to S. aureus, and which would eventually cause the cellular autolysis.”
(3) Lines 98–104 and 127–130, Figure 2a: Azalomycin F did not demonstrate a concentration-dependent release of LTA; the amounts of LTA liberated at 4 and 8 µg/mL are similar in quantity. This might suggest that Azalomycin F does not have a direct relation to LTA release. Please provide comments on this or make appropriate changes to the manuscript.
Additionally, for lines 111–113, conducting a kinetic experiment using a biochemical assay for membrane integrity along with LTA release will shed more light on the relationship between LTA release and membrane damage, determining whether they are exclusive events or interdependent.
Response 3: Thank you very much for your help and valuable suggestion for improving our work! We had inserted the information you suggested at Lines 119 to 121 in the revised manuscript. Simultaneously, we had provided some interpretation for this, and revised the related expressions, for your consideration. Please find them at Lines 121 to 122, 177 to 179, and 113 to 117.
You are right. Using a biochemical assay, the kinetic experiment for membrane integrity along with LTA release can shed more light on the relationship between LTA release and membrane damage. We had provided this information at Lines at 177 to 179. Thank you very much for your good and valuable suggestion to determine whether they are exclusive events or interdependent. We will improve our subsequent work for azalomycin F.
(4) Lines 181–197: Please provide data and refer to the figure number for all expression level data. Additionally, if the authors verify the expression level data using alternative techniques such as RT-PCR, western blot, etc., it will enhance their hypothesis.
Response 4: Thank you very much for your valuable suggestion and careful review! Using the parallel reaction monitoring (PRM) technology, we had further verified the expression level of six proteins, and provided these data and refer to the figure number as Figure S3 in the revised supplementary files. You are right, the expression level data from RT-PCR, western blot, etc. will enhance our hypothesis. Thank you very much for your help to improve our work!
(5) Lines 278–280, figure S3: The concentration of enzyme used for this assay is not mentioned.
Response 4: Thank you very much for your careful review and kind reminder! The concentrations of eLtaS (2.0 μM), LtaS (0.75 μM), and LtaS-embedded liposome (0.75 μM) used for the assay had been inserted in the caption of Figure S4 in the revised manuscript.
(6) Lines 319-321, figure 4e: Please put no enzyme control in this figure.
Response 6: Thank you very much for your careful review and valuable suggestion! You are right! A blank control without enzyme in Figure 4e should be set when the influence of azalomycin F on the LTA synthesis from eLtaS was explored, while this control was not set when we performed this experiment. Fortunately, no obvious change in DAG production under the azalomycin F concentrations of 0 and 10 μM (Figure 4e) indirectly indicated that azalomycin F in the control without enzyme likely cannot produce detectable DAG, and namely has little influence on the DAG production in this research.
(7) Section “2.5. Binding site of Azalomycin F to LtaS," Table S1: In this table, all tested amino acids show interaction with Azalomycin F; therefore, a negative control where Azalomycin F did not demonstrate any interaction with amino acids is required.
Response 7: Thank you very much for your careful review! As an auxiliary evidence, mass spectrometry mainly provides the relative strength of azamycin F binding amino acid residues for obtaining which amino acid residues azamycin F preferentially binds to in actual environments. Generally, the larger the intensity of complex ion from azalomycin F binding amino acids in MS analysis, the greater the possibility of their actual binding. Namely, we not only need explore which amino acids azalomycin F can bind to, more important, we need focus on which ones the intensities of their complex ions are larger. From this aspect, although Table S1 showed Azalomycin F can interact with all tested amino acids, the complex ion with smaller intensity suggested that the possibility of azalomycin F interacting with the amino acid is very less, such as AZF4a/Tyrosine or DMAZF4a/Tyrosine. Simultaneously, it is difficult to set a negative control that no complex ion was detected from the incubation mixture of azalomycin F and amino acids at this analytical condition since MS analysis is a highly sensitive method. So, it perhaps is not a very necessary to set a negative control in the MS analysis.
(8) Materials and Methods section: Please mention the manufacturer of the ELISA kit for LTA and diacylglycerol.
Response 8: Thank you very much for your careful review and kind reminder! As both sellers are the manufacturers, we had revised the text expression and thereout provided the manufacturers of the ELISA kits for LTA and diacylglycerol.
Other revision:
We had carefully performed extensive revision throughout the manuscript including all section and references, the spelling, formatting, syntax, linguistic edit and expressions, for your consideration.

Reviewer 2 Report
Comments and Suggestions for Authors
The manuscript describes mechanism of antimicrobial action of azalomycin F. It is well written paper which I recommend for publication after minor revisions.
Below there are specific comments/suggestions for the manuscript improvement.
Title: replace “from” by “of”.
The numbers of replicates and independent experiments should clearly be indicated for research methods used.
Provide names (or at least chemical classes) of compounds inhibiting lipoteichoic acid synthase mentioned as numbers e.g. in lines 55 and 56.
Indicate origin of all chemicals and reagents mentioned in the whole Materials and Methods section (e.g. in lines 867 and 868). Provide uniform information on suppliers e.g. BCA protein assay kit (Thermo Fisher Scientific, Waltham, MA, USA) and EASY‐nLC 1000 UPLC system (Thermo Fisher Scientific, San Jose, California).
Line 107: Change “S. Aureus” to “S. aureus”.
Line 352: Change “Azalomycin F” to “azalomycin F”.
Lines 655 and 656: Specify method (and provide reference) used for MIC determination.
The practical impact of the research should be discussed and summarized in the Discussion and/or Conclusion sections.
Comments on the Quality of English LanguageAlthough the English is fully understandable, the text needs some grammatical and typographical corrections.
Author Response
Dear Reviewer,
My co-authors and I are very grateful to you for your careful review, good comments, kind reminder and valuable suggestions. We have amended the manuscript according to the issues raised by you, and have pleasure to submit the revised version, together with the responses to your comments, for your consideration.
Many thanks for your kind attention!
Yours sincerely,
Ganjun Yuan
Here are our answers to your comments.
Comments and Suggestions for Authors
The manuscript describes mechanism of antimicrobial action of azalomycin F. It is well written paper which I recommend for publication after minor revisions.
Thank you for your careful review, good comments, and enthusiastic help to improve our work!
Below there are specific comments/suggestions for the manuscript improvement.
(1) Title: replace “from” by “of”.
Response 1: Thank you very much! We had revised it.
(2) The numbers of replicates and independent experiments should clearly be indicated for research methods used.
Response 2: Thank you very much for your valuable suggestion! We had inserted the numbers of replicates and independent experiments for the research method. As a supplement, this information was also indicated on the Figures and Tables in the result section.
(3) Provide names (or at least chemical classes) of compounds inhibiting lipoteichoic acid synthase mentioned as numbers e.g. in lines 55 and 56.
Response 3: Thank you very much for your valuable suggestion! We had inserted the chemical names of both compounds at Lines 58 and 59 of the revised manuscript.
(4) Indicate origin of all chemicals and reagents mentioned in the whole Materials and Methods section (e.g. in lines 867 and 868). Provide uniform information on suppliers e.g. BCA protein assay kit (Thermo Fisher Scientific, Waltham, MA, USA) and EASY‐nLC 1000 UPLC system (Thermo Fisher Scientific, San Jose, California).
Response 4: Thank you very much for your valuable suggestion! We had added a paragraph for providing the origin of more chemicals and reagents in subsection 3.1 of the Materials and Methods section (Lines 705 to 712). For important reagents, such as PG, DPPG, IPTG, and amino acids used for Mass analyses at Lines 867 and 868, we had provided uniform information on suppliers according to your suggestion.
(5) Line 107: Change “S. Aureus” to “S. aureus”.
Response 5: Thank you very much! We had revised it.
(6) Line 352: Change “Azalomycin F” to “azalomycin F”.
Response 6: Thank you very much! We had revised it.
(7) Lines 655 and 656: Specify method (and provide reference) used for MIC determination.
Response 7: Thank you very much for your valuable suggestion! We had specified the method used for MIC determination, and provide the reference at Line 696 of the revised manuscript.
(8) The practical impact of the research should be discussed and summarized in the Discussion and/or Conclusion sections.
Response 8: Thank you very much for your valuable suggestion and enthusiastic help to improve our work! We had inserted the practical impact of the research at Lines 680 to 688 in the Result and Discussion section of the revised manuscript, and at Lines 966 to 969 in the Conclusion section, for your consideration.
(9) Comments on the Quality of English Language
Although the English is fully understandable, the text needs some grammatical and typographical corrections.
Response 9: Thank you very much for your valuable suggestion! We had carefully performed extensive revision throughout the manuscript including the references and supplementary files, such as the compose type, spelling, formatting, syntax, linguistic edit and expression.

Reviewer 3 Report
Comments and Suggestions for Authors
The paper presented by Luo et al. reveals a novel mechanism of action for Azalomycin F as an LtaS inhibitor—a highly interesting target for antimicrobial compounds. The paper is well-structured, and the results substantiate both the initial hypothesis and the conclusions. In summary, the paper requires only a few changes and the repetition of one experiment before it can be published.
A major suggestion is for the authors to repeat the experiment presented in Figure 4e and i, or at the very least, modify the presentation of the results. Additionally, some of the explanations should be revised to enhance the overall understanding of the experiments' general objectives (further details are provided below). Also I suggest the authors to mix results and discussion in one section.
Introduction:
Derivative compounds from 1771 have also been recently published as LtaS inhibitors (PMCID: PMC10644342). Furthermore, other inhibitors of LTA synthesis have been reported (PMID: 35170959), and this information should be included in the introduction or discussion.
Section 2.1 In Figure 2, I recommend that the authors use keys to clarify the symbols being compared; otherwise, it becomes challenging to understand the results. Perhaps, the use of columns could enhance the comprehension of the statistics conducted. Along the same lines, differences between treatments mentioned in line 102 need clarification. Upon examining the figure, it appears that the group at 4 ug/mL shows significant differences at minutes 120 and 150 but not at 60, as the authors assert in the text.
Section 2.2
The authors chose 1.2 as the desired threshold. Therefore, I recommend that the authors provide some rationale for this decision.
In Figure S2, I suggest the authors change the colors for a more intuitive understanding, as green typically signifies upregulation and red indicates downregulation.
Since the experiments were conducted with n = 3, I propose that the authors provide the standard deviation of the results. This is especially important for cases where the upregulation results are so low that understanding their significance in bacterial behavior becomes challenging.
I recommend that the authors include a figure depicting the results of the genes involved in LTA function, showing their fold change and importance for LTA. This figure could be incorporated into Figure 2.
Regarding lines 247 to 251, the conclusions appear too speculative and are challenging to comprehend based on the results presented up to that point. It might be more appropriate to move this content to a discussion section. Furthermore, it is unclear how the authors establish a connection between the decrease of LTA and the upregulation of the proteins engaged in LTA biosynthesis, aside from the fact that Azalomycin F is an LtaS inhibitor.
In Figure 3, additional information in the footnotes should be included, as some of the axes on the graphs are unclear.
Lines 261 to 266 could be placed in the Section 2.3 of results or in the discussion.
Section 2.3.
Line 273: I would like to inquire about the authors' quantification method, as the given range (3 to 7 mg/mL) is too broad. I am interested in understanding how they quantified the protein before conducting the experiments to ensure accuracy in the amount added.
Lines 278 – 280: The authors should define what constitutes good activity in comparison with other reported activities.
Line 282: The authors should explain the meaning of DAG (diacylglycerol) since this acronym appears for the first time in the manuscript.
Figure 4: I suggest that the authors modify the title of the figure ("Influence of azalomycin F on the LTA synthesis from LtaS") in the footnotes as it does not accurately represent the figure.
Figure 4b: The footnote for part b is not clear. I recommend that the authors refine the explanation. Additionally, I suggest presenting a gel with a similar pattern as presented for eLtaS.
Figure 4e: I recommend that the authors change the graphs to inhibition curves showing the IC50 of the compound. Also, since the results are inconsistent in terms of significance, I suggest repeating the experiment. This section is crucial, and authors are encouraged to repeat the experiment and present the data as explained above.
Figure 4i: This result is difficult to understand. I propose that the authors rewrite this part of the paper, providing a clearer explanation for the obtained results. Additionally, the differences are not apparent, so an additional replicate may be necessary.
Line 329: Change the format of reference 22.
The use of liposomes is still not clearly explained, and the explanation given around line 330 is unclear. Therefore, the rationale should be better explained, as these results will be discussed throughout the paper.
Section 2.4:
Authors should introduce what DPPG is, along with the importance of this phospholipid as a model membrane for the assay. This information is crucial for readers to comprehend the context; otherwise, it could be challenging for some readers to follow.
Moreover, it is unclear why Azalomycin F can interact with LtaS-embedded liposomes, but it doesn't affect the activity in Figure 4i.
Section 2.5:
Line 469-470: I suggest that the authors add some information about the amino acids in the active center of LtaS when they are discussing it for the first time.
Material and Methods section:
Line 678: Add "Muller Hinton broth" before the acronym MHB.
Line 776: Add a space between "50" and "ug."
I suggest that the authors consistently use "ºC" instead of "K" for temperature units to maintain consistency.
The statistical analysis should be refined since the authors employed parametric analysis without a previous normality test. I recommend that the authors perform normality and homoscedasticity tests to determine whether parametric statistics are appropriate.
Authors should add information about MIC (Minimum Inhibitory Concentration) calculation or provide a reference if MIC was not analyzed. If MIC was not analyzed, it is highly recommended to ensure that they are working under the same conditions as other authors.
General minor mistakes:
Review the format of amino acid positions, deciding whether to use or not use spaces consistently throughout the paper.
The authors should provide results in molarity to facilitate comparison with results from other research works using different antimicrobials.
Line 106: Typo – "azalomyin" should be corrected to "azalomycin."
Comments on the Quality of English LanguageI have found several mistakes and some parts are hard to understand, please review the english of the paper in general.
Author Response
Dear Reviewer,
My co-authors and I are very grateful to you for your careful review, good comments, kind reminder, valuable suggestions, and lots of effort for improving our work. We have amended the manuscript according to the issues raised by you, and have pleasure to submit the revised version, together with the responses to your comments, for your consideration.
Many thanks for your kind attention!
Yours sincerely,
Ganjun Yuan
Here are our answers to your comments.
Comments and Suggestions for Authors
(1) The paper presented by Luo et al. reveals a novel mechanism of action for Azalomycin F as an LtaS inhibitor—a highly interesting target for antimicrobial compounds. The paper is well-structured, and the results substantiate both the initial hypothesis and the conclusions. In summary, the paper requires only a few changes and the repetition of one experiment before it can be published.
Response 1: Thank you for your careful review, good comments, and enthusiastic help to improve our work! We had extensively revised the manuscript.
(2) A major suggestion is for the authors to repeat the experiment presented in Figure 4e and i, or at the very least, modify the presentation of the results. Additionally, some of the explanations should be revised to enhance the overall understanding of the experiments' general objectives (further details are provided below). Also I suggest the authors to mix results and discussion in one section.
Response 2: Thank you for your careful review, valuable suggestions, and lots of effort for improving our work. According to your suggestion, we had mixed results and discussion in one section, and provided many explanations for enhancing the overall understanding of the experiments' general objectives. You are right! It is better to repeat the experiment or at least modify the presentation of the results for Figure 4e and i. However, we found it is difficult for us to complete it. So, we provided a crucial fact for supporting the related conclusion before we started this research, and many interpretations and our consideration for the reason of no change given to Figure 4e and I. The fact together with related interpretations had been inserted at Lines 354 to 360, 378 to 382, and 387 to 390 of the manuscript. Simultaneously, the detailed explanations were provided in the following responses, for your consideration.
Introduction:
(3) Derivative compounds from 1771 have also been recently published as LtaS inhibitors (PMCID: PMC10644342). Furthermore, other inhibitors of LTA synthesis have been reported (PMID: 35170959), and this information should be included in the introduction or discussion.
Response 3: Thank you for your valuable and important information to improve our work! We had downloaded both papers and carefully read it. Depending on the conclusions of both papers, we had provided this information at Lines 65 to 70 of the introduction section and at Lines 677 to 679 of the results and discussion one for your consideration.
(4) Section 2.1 In Figure 2, I recommend that the authors use keys to clarify the symbols being compared; otherwise, it becomes challenging to understand the results. Perhaps, the use of columns could enhance the comprehension of the statistics conducted. Along the same lines, differences between treatments mentioned in line 102 need clarification. Upon examining the figure, it appears that the group at 4 ug/mL shows significant differences at minutes 120 and 150 but not at 60, as the authors assert in the text.
Response 4: Thank you for your careful review and kind reminder! We had carefully checked them. As we had initially wanted to present the significant difference between 4 (or 8) μg/mL and 1 (or 2) μg/mL, symbols # and ##, and ! and !! were used in Figure 2. However, the text didn’t mention them, and these become challenging to understand the results. Thereby, we deleted these symbols on Figure 2 of the revised manuscript for easily understanding the results. Reminded by you, we had carefully checked the significant differences for 4 and 8 ug/mL groups on Figure 2 and the corresponding text expression at Lines 102 of the original manuscript, and found some errors in the text expression. So, we had revised them and provided some interpretations for the results at Lines 109 to 122 in the revised manuscript.
Section 2.2
(5) The authors chose 1.2 as the desired threshold. Therefore, I recommend that the authors provide some rationale for this decision.
Response 5: Thank you for your good question! Generally, the magnitude and significance of the differences between groups are our focus on when analyzing the experimental results. From a statistical point of view, differences can arise from the intra-group errors and the inter-group substantial differences. Therefore, the first thing to consider is whether the experimental differences come from inter-group substantive ones or be caused by experimental errors. If the difference is caused by experimental errors, no matter how large the difference is, it is not an inter-group substantive one, and cannot be considered that there is a significant difference between the two groups. Conversely, even the small difference, if they are caused by inter-group substantial ones rather than experimental errors, it can also indicate significant differences between the two groups. According to statistics, whether the difference is an inter-group substantial difference or caused by experimental errors depends on the P-value. If the P-value is less than 0.05 (or 0.01), the difference comes from inter-group substantial differences. Otherwise, it may be caused by the experimental errors. Based on this, the threshold requirement for the fold change can be large or small. Namely, all the fold change of 1.2, 1.3, 1.5, or 2.0 is acceptable. Of course, the lager the fold change, the better. If the calculated P-value after selecting the threshold of fold change is less than 0.05, it means that this difference comes from the inter-group substantive differences and is not caused by the experimental errors. Otherwise, it is caused by intra-group errors. Based on the above statistical viewpoint, we have chosen 1.2 as the threshold of fold change, but its corresponding P-value must be less than 0.05.
(6) In Figure S2, I suggest the authors change the colors for a more intuitive understanding, as green typically signifies upregulation and red indicates downregulation.
Response 6: Thank you for kind reminder and valuable suggestion! You are right! Green typically signifies upregulation and red indicates downregulation. Another, we also found many papers presented the KEGG pathway in the same color to our presentation. Simultaneously, we had provided notes for these colors. So, we are sorry for that we kept the color unchanged in Figure S2.
(7) Since the experiments were conducted with n = 3, I propose that the authors provide the standard deviation of the results. This is especially important for cases where the upregulation results are so low that understanding their significance in bacterial behavior becomes challenging.
Response 7: Thank you for your careful review and valuable suggestion! We had performed the verification quantification for targeted proteins using the parallel reaction monitoring (PRM) technology. Thereby, we had provided the detailed original data of the quantitative information of six proteins in azalomycin F and blank groups, and shown as Figure S3 in the updated supplementary files together with their reference numbers, for your consideration.
(8) I recommend that the authors include a figure depicting the results of the genes involved in LTA function, showing their fold change and importance for LTA. This figure could be incorporated into Figure 2.
Response 8: Thank you for your valuable suggestion! In the manuscript, six proteins related to the cell envelope were quantified using PRM technology. Among them, three proteins as lipoteichoic acid synthase (Q2G093), signal peptidase I (Q2FZT7) and teichoic acids export ATP-binding protein TagH (Q2G2L1) involves the LTA biosynthesis and function, while we feel all these three proteins are important for LTA biosynthesis and function. Simultaneously, the genes involved in LTA function remains completely unknown, and their fold change can be easily presented in the text. Simultaneously, their quantitative data were added as Figure S3 in the updated supplementary files. So, we are sorry for that we didn’t add a figure in Figure 3.
(9) Regarding lines 247 to 251, the conclusions appear too speculative and are challenging to comprehend based on the results presented up to that point. It might be more appropriate to move this content to a discussion section. Furthermore, it is unclear how the authors establish a connection between the decrease of LTA and the upregulation of the proteins engaged in LTA biosynthesis, aside from the fact that Azalomycin F is an LtaS inhibitor.
Response 9: Thank you for your valuable suggestion! You are right! The conclusions at Lines 247 to 251 appear too speculative. For supporting this conclusion, we had used the result from proteomics at once in the next two sentences (Lines 275 to 280 of the revised manuscript). As you pointed out, the conclusion is challenging to comprehend based on the results presented up to that point since the conclusion of azalomycin F inhibiting LTA biosynthesis was concluded from the next section (section 2.3). Thereout, we had revised the expression and provided some additional explanations. Simultaneously, we combined the result section and discussion one for more convenient expression according to your suggestion.
Furthermore, one of our aims is to conclude that azalomycin F can disrupt the cell envelope of S. aureus, by accelerating the LTA release, damaging the cell membrane, and other possible mechanism which will feedback upregulate the proteins stabling the cell envelope. Another is to prove that azalomycin F can inhibit LtaS to synthesize LTA. So, we did not establish a connection between the decrease of LTA and the upregulation of the proteins engaged in LTA biosynthesis, while only provided possible inferences from the perspective of biofeedback.
(10) In Figure 3, additional information in the footnotes should be included, as some of the axes on the graphs are unclear.
Response 10: Thank you for your valuable suggestion! Some interpretations were unintuitively presented for the figure according to the general expression form of bioinformatics analysis, while not in the footnotes. Reminded by you, we had revised Figure 3 and provided some information of the axes in a more intuitive way although they are different from general x- or y- axis form.
(11) Lines 261 to 266 could be placed in the Section 2.3 of results or in the discussion.
Response 11: Thank you for your valuable suggestion! You are right! It is more appropriate to move this content (Lines 261 to 266) to the Section 2.3 of results or the discussion. Since sections result and discussion were combined, we placed this content at Lines 391 to 396 in the section 2.3 of the revised manuscript.
Section 2.3.
(12) Line 273: I would like to inquire about the authors' quantification method, as the given range (3 to 7 mg/mL) is too broad. I am interested in understanding how they quantified the protein before conducting the experiments to ensure accuracy in the amount added.
Response 12: Thank you for your good question and kind reminder! The concentrations of purified eLtaS were determined using a modified BCA protein assay kit. Generally, the concentrations of eLtaS prepared will fall into the range from 3 to 7 mg/mL according our procedure. Before we used the eLtaS solution for conducting the experiments, we will determine the accurate concentration using a modified BCA protein assay kit. Depended on the accurate concentration, we added the accurate amount. We sincerely apologize for the inconvenience caused to your understanding due to our vague expression. Thereby, we had revised the expression.
(13) Lines 278 – 280: The authors should define what constitutes good activity in comparison with other reported activities.
Response 13: Thank you for your kind reminder! We sincerely apologize for the inconvenience caused to your understanding due to our inaccurate expression. Thereby, we had revised the expression at Line 297 in the revised manuscript.
(14) Line 282: The authors should explain the meaning of DAG (diacylglycerol) since this acronym appears for the first time in the manuscript.
Response 14: Thank you for your careful review and valuable suggestion! We had provided the full name for DAG at Line 297 when this acronym appears for the first.
(15) Figure 4: I suggest that the authors modify the title of the figure ("Influence of azalomycin F on the LTA synthesis from LtaS") in the footnotes as it does not accurately represent the figure.
Response 15: Thank you for your kind reminder and valuable suggestion! We had revised it as “Preparations and analyses of eLtaS and LtaS-embedded liposomes, and the influence of azalomycin F on both enzyme activities” for your consideration.
(16) Figure 4b: The footnote for part b is not clear. I recommend that the authors refine the explanation. Additionally, I suggest presenting a gel with a similar pattern as presented for eLtaS.
Response 16: Thank you for your kind reminder and valuable suggestion! After we had checked it, we found that the lane marks were misaligned. So, we aligned the lane marks. Another, we refined the ambiguous explanation according to your suggestion. However, we did not present the gel with a similar pattern as presented for eLtaS for displaying the actual experimental process.
(17) Figure 4e: I recommend that the authors change the graphs to inhibition curves showing the IC50 of the compound. Also, since the results are inconsistent in terms of significance, I suggest repeating the experiment. This section is crucial, and authors are encouraged to repeat the experiment and present the data as explained above.
Response 17: Thank you for your valuable suggestion! You are right! When we did it according your suggestion, we found that it was difficult to showing the IC50 of azalomycin F inhibiting on eLtaS and LtaS since the IC50 can be not calculated from the range of experimental azalomycin F concentrations. You are right, this section is crucial for the conclusion on azalomycin F inhibiting the LTA synthesis from eLtaS. However, it is difficult for our lab to repeat the experiment in a short time during the holidays and festivals. Another, as Figure 4e showed, it indicated that azalomycin F can obviously inhibit the enzyme activities of eLtaS, especially for 80 and 160 μM groups, although other concentrations of azalomycin F did not present the consistent significance due to the large errors for intra-groups in the experiments.
More importantly, a crucial fact that azalomycin F-induced S. aureus lysis can be prevented by cellular LTA was confirmed by us (Yuan, et al., Biomedicine & Pharmacotherapy 2019, 109: 1940–1950) before this experiment was developed. Considering this fact, here the reliability of the results from this section should be enhanced although only two groups presented significant (P<0.05) and very significant (P<0.01) difference in statistics. Simultaneously, some compounds like compound 1171 and its derivatives at first reported as LtaS inhibitor in vitro was eventually proved that they are not LtaS inhibitors except for Congo red (Douglas, et al., ACS Infect. Dis. 2023, 9, 2141−2159), since neither deletion nor overexpression of LtaS altered the susceptibility of S. aureus to them. Thereout, we did not repeat the experiment, while had improved the text of this section and provided above related information and interpretation at Lines 351 to 360, for your consideration.
(18) Figure 4i: This result is difficult to understand. I propose that the authors rewrite this part of the paper, providing a clearer explanation for the obtained results. Additionally, the differences are not apparent, so an additional replicate may be necessary.
Response 18: Thank you for your kind reminder and valuable suggestion! According to your suggestion, we had rewritten this part of the paper and provided a clearer explanation for the obtained results at Lines 378 to 390. As you pointed out, the differences are not apparent. This might be due to that the actual concentrations of azalomycin F acting on the LtaS embedded in the liposome were largely reduced from the molecular consumption of azalomycin F binding to the phospholipid polar head of the liposome. Simultaneously, during the preparation and incubation of LtaS-embedded liposome, some residual LtaS unembedded and possible eLtaS degraded from LtaS might be also responsible for this result since the eLtaS and LtaS unembedded in the liposome can also catalyze PG to synthesize the LTA.
As a membrane proteins, the expression level of LtaS was low, and the enzyme was easily degradable during its preparation. So, it is difficult to obtain high concentraions of LtaS embedded in the liposome. Simultaneously, azalomycin F will quickly bind to the phospholipid polarity head of LtaS-embedded liposome in this incubation system, and the actual concentrations of azalomycin F interacting with LtaS will largely decrease. Thereby, many factors will affect LtaS on the synthesis of LTA in the incubation system. We had tried to obtain the results with significant differences. However, we failed to achieve it except that the downward trend could be observed after the incubation system interfered by the concentrations of azalomycin F from 80 to 160 μM. Considering that the prevention of azalomycin F-induced S. aureus lysis by the cellular LTA and the inhibition of azalomycin F on eLtaS to synthesize the LTA, it is our another aim to provide Figure 4i and related data as a reference in research design for LtaS-related researches since there has been no exploration for compounds inhibiting the enzyme activity of LtaS in similar incubation systems so far.
(19) Line 329: Change the format of reference 22.
Response 19: Thank you for your careful review and kind reminder! We had already revised it.
(20) The use of liposomes is still not clearly explained, and the explanation given around line 330 is unclear. Therefore, the rationale should be better explained, as these results will be discussed throughout the paper.
Response 20: Thank you for your careful review and valuable suggestion! We had provided a clear explanation for the use of liposomes at Lines 361 to 363 in the revised manuscript, also shown as “Since LtaS is a transmembrane protein, LtaS-embedded liposome was further pre-pared to simulate the physiological environment of the LTA biosynthesis in S. aureus as far as possible, using the cell membrane phospholipids extracted from S. aureus according to the reported method [22] and analyzed by thin layer chromatography (TLC).” for your consideration.
Section 2.4:
(21) Authors should introduce what DPPG is, along with the importance of this phospholipid as a model membrane for the assay. This information is crucial for readers to comprehend the context; otherwise, it could be challenging for some readers to follow.
Response 21: Thank you for your valuable suggestion! We had added some introduction about DPPG at Lines 405 to 409, also shown as follows: “It was worth noting that 1,2-dihexadecanoyl-sn-glycero-3-phospho-(1'-rac-glycerol) (DPPG) is a main component of cell-membrane phospholipids of S. aureus and here was mainly used as a component of control systems (B) and (C) relative to systems (D) and (E) contained LtaS-embedded liposomes.”
(22) Moreover, it is unclear why Azalomycin F can interact with LtaS-embedded liposomes, but it doesn't affect the activity in Figure 4i.
Response 22: Thank you for your careful review and inspiring questions! The enzyme activity of LtaS embedded in the liposomes obviously enhanced by azalomycin F when which concentrations at 20 μM (P < 0.05) or 80 μM (P < 0.01) (Figure 4i), being relative to blank group (0 μM). However, there were no significant difference (P > 0.05) among various concentrations ranged from 10 to 160 μM of azalomycin F. For this case, we had provided some interpretation at Lines 382 to 390 in the manuscript, also shown as follows for your consideration: “this might be due to that the actual concentrations of azalomycin F acting on the LtaS embedded in the liposome were largely reduced from the molecular consumption of azalomycin F binding to the phospholipid polar head of the liposome [19]. Nonetheless, the downward trend could be observed after the incubation system interfered by the concentrations of azalomycin F from 80 to 160 μM. Simultaneously, during the preparation and incubation of LtaS-embedded liposome, some residual LtaS unembedded and possible eLtaS degraded from LtaS might be also responsible for this result since the eLtaS and LtaS unembedded in the liposome can also catalyze PG to synthesize the LTA.”
Section 2.5:
(23) Line 469-470: I suggest that the authors add some information about the amino acids in the active center of LtaS when they are discussing it for the first time.
Response 23: Thank you for your valuable suggestion! We had already inserted some information about the amino acids in the active center of LtaS at Lines 518 to 520 of the revised manuscript, also shown as follows: “Among thirteen important amino acid residues reported in the LtaS active center, residues Trp354 and Tyr477 respectively involve the binding of LtaS to the substrates and the stabilization of the growing LTA chain in the pocket [9].” for your consideration.
Material and Methods section:
(24) Line 678: Add "Muller Hinton broth" before the acronym MHB.
Response 24: Thank you for your careful review and kind reminder! We had already added it.
(25) Line 776: Add a space between "50" and "ug."
Response 25: Thank you for your careful review and kind reminder! We had already added it.
(26) I suggest that the authors consistently use "ºC" instead of "K" for temperature units to maintain consistency.
Response 26: Thank you for your careful review and good suggestion! We had already revised them according to your suggestion.
(27) The statistical analysis should be refined since the authors employed parametric analysis without a previous normality test. I recommend that the authors perform normality and homoscedasticity tests to determine whether parametric statistics are appropriate.
Response 27: Thank you for your kind reminder and valuable suggestion! We had performed normality and homoscedasticity tests using the software of Data Processing System (DPS, Zhejiang University, Hangzhou, China). Fortunately, all data presented normality and homoscedasticity, and all the difference results were consistent with the original ones, and so remained unchanged. Another, we had provided above information at Lines 948 to 951 of the revised manuscript.
(28) Authors should add information about MIC (Minimum Inhibitory Concentration) calculation or provide a reference if MIC was not analyzed. If MIC was not analyzed, it is highly recommended to ensure that they are working under the same conditions as other authors.
Response 28: Thank you for your careful review and kind reminder! Although the MIC of azalomycin F against S. aureus is known to us in our lab, we will make it sure again before a new study starts. The MIC was determined on 96-well plates, using broth microdilution method. According to your suggestion, we had already added the test method and a reference at Line 696 for your consideration.
General minor mistakes:
(29) Review the format of amino acid positions, deciding whether to use or not use spaces consistently throughout the paper.
Response 29: Thank you for your careful review and kind reminder! we had checked the format of amino acid positions throughout the paper, and found two amino acid positions to use spaces (Trp 354 and Ser 480). We had revised them as which not to use spaces (Trp354 and Ser480).
(30) The authors should provide results in molarity to facilitate comparison with results from other research works using different antimicrobials.
Response 30: You are right! Providing the results of molar concentration can facilitate to compare with the results from other research works using different antimicrobials. As the MIC was usually expressed as “μg/mL” according to the determination with broth microdilution method, sometimes we had provided the results in “μg/mL” for facilitating the comparison of used concentration with its MIC, such as in sections 2.1 and 2.2. However, sometimes we will also provide the results in molar concentration, and which depends on the research contents and the format generally used by other researchers, such as the interaction of azalomycin F with enzyme.
(31) Line 106: Typo – "azalomyin" should be corrected to "azalomycin."
Response 31: Thank you for your careful review and kind reminder! We had already revised it.
(32) Comments on the Quality of English Language
I have found several mistakes and some parts are hard to understand, please review the english of the paper in general.
Response 32: Thank you very much for your careful review and valuable suggestion! We had carefully performed extensive revision throughout the manuscript including the supplementary files and references, such as the spelling, formatting, syntax, linguistic edit and expression.

Round 2
Reviewer 1 Report
Comments and Suggestions for Authors
Authors have provided satisfactory explanations for most of the comments. However, I have one additional concern regarding Section 2.5, binding site of azalomycin F to LtaS. The authors did not conduct an Azalomycin F binding experiment with amino acids using mass spectrometry with any negative control, and they did not analyze the interaction of Azalomycin F with respective amino acid mutants of LtaS. As a result, the analysis of the amino acids involved in the Azalomycin F interaction appears to be speculative. I recommend that the authors include appropriate comments addressing this limitation in their manuscript.
Author Response
Dear Reviewer,
My co-authors and I are very grateful to you for your careful review, good comments, valuable suggestions, and lots of effort for improving our work! We have amended the manuscript according to the issues raised by you, and have pleasure to submit the revised version, together with the responses to your comments, for your consideration.
Many thanks for your kind attention!
Yours sincerely,
Ganjun Yuan
Here are our answers to your comments.
Comments and Suggestions for Authors
Authors have provided satisfactory explanations for most of the comments. However, I have one additional concern regarding Section 2.5, binding site of azalomycin F to LtaS. The authors did not conduct an Azalomycin F binding experiment with amino acids using mass spectrometry with any negative control, and they did not analyze the interaction of Azalomycin F with respective amino acid mutants of LtaS. As a result, the analysis of the amino acids involved in the Azalomycin F interaction appears to be speculative. I recommend that the authors include appropriate comments addressing this limitation in their manuscript.
Response 1: Thank you for your kindness, good comments, valuable suggestions, and warm help for improving our work! According to your suggestion, we had inserted some discussion about this at Lines 562 to 567 of the revised manuscript, as follows: “As the MS analyses for azalomycin F binding various amino acids of LtaS active center were performed without negative controls, the results needed to require more supports from synchronous fluorescence spectra and molecular docking. Another, it would further improve the reliability of these results if the analyses for the interactions of azlomycin F with various amino acid mutants of LtaS were performed based on these results.” for your consideration. Learning from you, we will further improve our work in the subsequent research.
Other revision:
We had carefully performed extensive revision throughout the manuscript including the supplementary file and references once again, such as the spelling, formatting, syntax, linguistic edit and expressions, for your consideration.

Reviewer 3 Report
Comments and Suggestions for Authors
Dear authors, thank you for providing a new version of the manuscript. You have improved the quality of the paper's presentation, and I believe it is ready to be published. I have identified minor English mistakes, but it is just a proofreading task. The only issue found in the revised version was the absence of new experiments confirming some of the previous results. However, I understand that the paper is of high quality, and the deviations are not significant, so it could probably be published without further comments
Comments on the Quality of English LanguageSome minor mistakes have been detected in the revised version of the manuscript.
Author Response
Dear Reviewer,
My co-authors and I are very grateful to you for your careful review, good comments, valuable suggestions, warm help, and lots of effort for improving our work. We have amended the manuscript according to the issues raised by you, and have pleasure to submit the revised version, together with the responses to your comments, for your consideration.
Many thanks for your kind attention!
Yours sincerely,
Ganjun Yuan
Here are our answers to your comments.
Comments and Suggestions for Authors
Dear authors, thank you for providing a new version of the manuscript. You have improved the quality of the paper's presentation, and I believe it is ready to be published. I have identified minor English mistakes, but it is just a proofreading task. The only issue found in the revised version was the absence of new experiments confirming some of the previous results. However, I understand that the paper is of high quality, and the deviations are not significant, so it could probably be published without further comments.
Response 1: Thank you for your kindness, careful review, good comments, valuable suggestions, warm help, and lots of effort for improving our work! Learning from you, we will further improve our work in the subsequent research.
Comments on the Quality of English Language
Some minor mistakes have been detected in the revised version of the manuscript.
Response 2: Thank you very much for your careful review and warm help! We had carefully performed extensive revision throughout the manuscript including the supplementary files and references once again, such as the spelling, formatting, syntax, linguistic edit and expression, for your consideration.
